# Highly efficient nonlinear optical emission from a subwavelength crystalline silicon cuboid mediated by supercavity mode

Mingcheng Panmai[1,4], Jin Xiang [1,4], Shulei Li [1], Xiaobing He[1], Yuhao Ren[2], Miaoxuan Zeng[3], Juncong She [3], Juntao Li [2✉] & Sheng Lan [1✉]

The low quantum efficiency of silicon (Si) has been a long-standing challenge for scientists. Although improvement of quantum efficiency has been achieved in porous Si or Si quantum dots, highly efficient Si-based light sources prepared by using the current fabrication technnoloy of Si chips are still being pursued. Here, we proposed a strategy, which exploits the intrinsic excitation of carriers at high temperatures, to modify the carrier dynamics in Si nanoparticles. We designed a Si/SiO$_2$ cuboid supporting a quasi-bound state in the continuum (quasi-BIC) and demonstrated the injection of dense electron-hole plasma via two-photon-induced absorption by resonantly exciting the quasi-BIC with femtosecond laser pulses. We observed a significant improvement in quantum efficiency by six orders of magnitude to ~13%, which is manifested in the ultra-bright hot electron luminescence emitted from the Si/SiO$_2$ cuboid. We revealed that femtosecond laser light with transverse electric polarization (i.e., the electric field perpendicular to the length of a Si/SiO$_2$ cuboid) is more efficient for generating hot electron luminescence in Si/SiO$_2$ cuboids as compared with that of transverse magnetic polarization (i.e., the magnetic field perpendicular to the length of a Si/SiO$_2$ cuboid). Our findings pave the way for realizing on-chip nanoscale Si light sources for photonic integrated circuits and open a new avenue for manipulating the luminescence properties of semiconductors with indirect bandgaps.

[1] Guangdong Provincial Key Laboratory of Nanophotonic Functional Materials and Devices, School of Information and Optoelectronic Science and Engineering, South China Normal University, 510006 Guangzhou, People's Republic of China. [2] State Key Laboratory of Optoelectronic Materials and Technologies, School of Physics, Sun Yat-sen University, 510275 Guangzhou, People's Republic of China. [3] State Key Laboratory of Optoelectronic Materials and Technologies, Guangdong Province Key Laboratory of Display Material and Technology, School of Electronics and Information Technology, Sun Yat-sen University, 510275 Guangzhou, People's Republic of China. [4] These authors contributed equally: Mingcheng Panmai, Jin Xiang. ✉email: lijt3@mail.sysu.edu.cn; slan@scnu.edu.cn

On-chip signal processing and optical computation require signal generation, transportation and detection. The photonic integrated circutes in the future will be constructed with light emitting devices, waveguides and detectors. As one of the most important semiconductors[1], silicon (Si) is the core element for electronic devices and a key material for optical waveguides and detectors[2–4]. Unfortunately, it is generally considered as poor photon emitter because of its indirect bandgap[1,5,6]. So far, a variety of strategys have been suggested to enhance the quantum efficiency of Si and most of them rely on band diagram engineering[7]. Quantum size effect and defect states are utilized to enhance the quantum efficiencies of Si quantum dots[8–12] and porous Si[13], respectively. In addition, efficient white light emission was also observed in Au/Si alloy nanoparticles fabricated by using laser ablation[14]. Very recently, highly efficient emission from hexagonal Ge and Si/Ge alloy was demonstrated by bandstructure engineering[15]. In comparison, less attention has been paid to light-matter interaction, which can be exploited to significantly modify the carrier dynamics in crystalline Si.

Owing to the large refractive index (~4.0) and small absorption of Si in the visible to near infrared spectral range, Si nanoparticles with appropriate sizes support Mie resonances and act as artificial atoms for metamaterials operating at optical frequencies[16]. The strongly localized electric fields at the Mie resonances of a Si nanoparticle render it greatly enhanced nonlinear optical responses[17]. Highly efficient harmonic generation and hot electron luminescence from single Si nanoparticles have been demonstrated by resonantly exciting the Mie resonances (e.g., the magnetic dipole (MD) resonance)[18,19]. More importantly, it is revealed that the carrier dynamics in a Si nanoparticle can be significantly changed by injecting high-density carriers[20]. As a result, the significantly enhanced Auger recombination process in combination with the accelerated radiative recombination processes mediated by the electric and magnetic quadrupole resonances increase the quantum efficiency by several orders of magnitude[18]. When the injected carrier density exceeds a critical value, it is expected that the radiative recombination rate will become proportional to the carrier density, resulting in a further enhancement in the quantum efficiency[21,22]. On the other hand, the high temperature in the Si nanoparticle resulting from the thermalization of hot carriers may induce the intrinsic excitation of carriers. In this case, high-density electrons can be generated at the bottom of the conduction band ($\Delta$ point), continuously supplying hot electrons for the interband radiative recombination mediated by phonons[22]. Therefore, how to generate dense electron-hole plasma in a Si nanoparticle has become the key point to produce highly efficient hot electron luminescence.

Previously, the enhanced the two-photon-induced absorption (TPA) achieved at the MD resonance of a Si nanoparticle and the mirror-image-induced MD of a particle-on-film system have been exploited to inject high-density carriers into the Si nanoparticle[18,21,22]. However, the quality (Q) factors of such optical modes (~10) are not large enough to fully exploit the TPA of the Si nanoparticle induced by femtosecond laser pulses of ~100 fs, which possesses a linewidth of ~10 nm in the near infrared spectral range (i.e., ~800 nm). From the viewpoint of spectral match, an optical mode with a Q factor of ~80 is necessary in order to efficiently inject carriers into the Si nanoparticle via a TPA process.

Recently, it has been shown theoretically and demonstrated experimentally that optical modes with extremely large Q factors can be achieved in the so-called bound states in the continuum (BICs)[23–25]. Basically, BICs with infinite Q factors can only be found in periodic structure with infinitely large sizes[26]. In practical applications, however, periodic structures with finite sizes are generally employed to achieve quasi-BICs with finite but huge

Q factors[27,28]. It was demonstrated that a Q factor as high as ~18,000 can be obtained in a regular array of Si nanoparticles of ~50 periods with symmetry breaking[29]. To match the narrow linewidth of the high-Q BIC, picosecond laser pulses were employed to excite the array of Si nanoparticles. For quasi-BICs with relatively low-Q factors, the match of linewidth can be realized by using femtosecond laser pulses, which generally possess a Q factor of ~80 in the near infrared spectral range. The resonant excitation of the quasi-BIC leads to an enhancement of the second harmonic generation from Si nanoparticles by several orders of magnitude[30,31]. In addition, it was shown that the photoluminescence from germanium (Ge) quantum dots embedded in Si matrix, which is employed as the light source at telecommunication wavelength, was enhanced by using photonic structures supporting BICs[32,33]. From the viewpoint of integration, a nanoscale photon emitter with significantly reduced volume is highly desirable[34,35]. Fortunately, quasi-BICs are also available in single Si nanoparticles by utilizing the interference between the Mie resonances, which lead to dramatically enhanced near-field intensity and significantly reduced far-field radiation[28,36–39].

In this article, we proposed the use of the quasi-BIC or supercavity mode supported by a Si/SiO$_2$ cuboid to inject dense electron-hole plasma and demonstrated ultra-bright hot electron luminescence from the Si/SiO$_2$ cuboid. It was revealed numeically and experimentally that a quasi-BIC can be established by making use of the coherent interaction between the electric dipole (ED), MD, and magnetic octupole (MO) modes supported by the Si/SiO$_2$ cuboid. A significant enhancement in TPA could be achieved at the quasi-BIC because of the match of its frequency spectrum with that of the femtosecond laser pulse. Burst of hot electron luminescence was observed when the excitation pulse energy exceeded a threshold, which depends strongly on the polarization of the laser light. A quantum efficiency as high as ~13% was achieved, implying an enhancement of more than six orders of magnitude as compared with the value for bulk Si. Our findings indicate the potential applications of Si nanoparticles in highly efficient white light sources and the possibility of realizing on-chip Si nanolasers with tunable wavelength.

## Results

**Geometry and structure of Si/SiO$_2$ cuboid.** Electron-beam lithography in combination with reactive ion etching were employed to fabricate the Si cuboids on a sapphire (Al$_2$O$_3$) substrate (see Methods and Supplementary Note 1). By deliberately designing the geometrical parameters of a Si cuboid, a quasi-BIC can be activated in the Si cuboid by using a transverse electric (TE) polarized wave (Fig. 1a). In order to eliminate nonradiative recombination centers introduced in the etching process, a passivation process was employed to create thin oxide layers on the surfaces of Si cuboids (Fig. 1b and Supplementary Note 1). For clarity, we use $l'$, $w'$, $h'$ to denote the length, width, and height of the inner Si cuboid and $l$, $w$, $h$ to denote the corresponding parameters of the Si/SiO$_2$ cuboid (Fig. 1b). Apparently, we have $l = l' + 2t$, $w = w' + 2t$, $h = h' + t$, where $t$ is the thickness of the outer SiO$_2$ layer. The morphologies of the fabricated Si/SiO$_2$ cuboids were examined by using scanning electron microscopy (SEM), from which the geometrical parameters of Si/SiO$_2$ cuboids (i.e., $l$, $w$, $h$) could be extracted (Fig. 1c and Supplementary Note 2).

**Physical mechanism for realizing efficient nonlinear optical emission from Si/SiO$_2$ cuboid.** In Fig. 1d, we illustrate the physical mechanism for realizing efficient nonlinear optical emission from Si/SiO$_2$ cuboids by exploiting the significantly

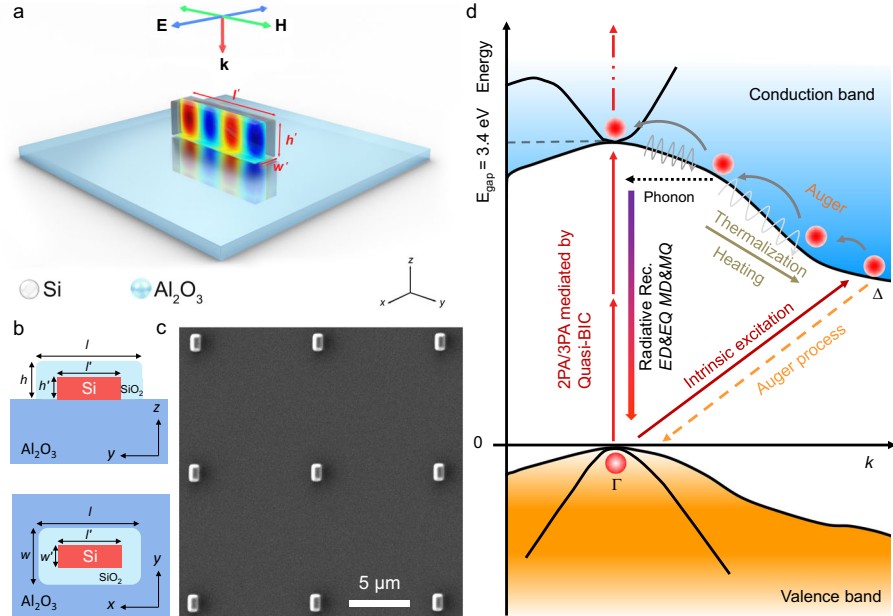

**Fig. 1 Structure and morphology of Si/SiO₂ cuboids supporting quasi-BICs and the physical mechanism for efficient white light emission. a** Schematic showing a Si cuboid supported by a sapphire (Al₂O₃) substrate. Also shown is the magnetic field distribution at the quasi-BIC. **b** Schematic showing the detailed structure of a Si/SiO₂ cuboid supported by a sapphire substrate, which is defined by geometrical parameters of $l$, $w$, $h$, $l'$, $w'$, $h'$, and $t$. **c** SEM image of a regular array of Si/SiO₂ cuboids. **d** Schematic showing the physical mechanism for efficient white light emission from Si/SiO₂ cuboids by exploiting the enhanced nonlinear optical absorption at quasi-BICs. Here, ED, EQ, MD, MQ represent electric dipole, electric quadrupole, magnetic dipole, and magnetic quadrupole, respectively. 2PA and 3PA represent two- and three-photon-induced absorption.

enhanced nonlinear optical absorption achieved at the quasi-BICs supported by Si/SiO₂ cuboids. Basically, high-density electron-hole pairs can be generated in a Si/SiO₂ cuboid via two- or three-photon-induced absorption (2PA or 3PA) upon the excitation of the Si/SiO₂ cuboid by using femtosecond laser pulses. Since the 2PA or 3PA is proportional to the fourth or sixth power of the electric field inside the Si cuboid (i.e., $|E|^4$ or $|E|^6$), a significantly enhanced 2PA or 3PA is expected at the quasi-BIC where maximum electric field (or Q factor) is achieved. In general, the hot electrons generated in the conduction band of Si will relax rapidly (less than 1.0 ps) from the Γ point to the Δ point via the emission of phonons and then recombine radiatively with the holes in the valence band with the help of phonons. However, such relaxation and recombination processes will be greatly alleviated in the high-density case by the Auger process, which is proportional to the cubic of the carrier density. As a result, the electrons remain "hot" at the high-energy states around the Γ point, which increases dramatically the relaxation time by two orders of magnitude (~50 ps) or the possibility for the vertical transition to the valence band. The Mie resonances of the Si/SiO₂ cuboid with emhanced electric field, such as electric and magnetic dipoles and quadrupoles (ED, MD, EQ, and MQ etc.), will accelerate the radiative recombination of hot electrons via the Purcell effect (~500 ps). Moreover, the radiative recombination time becomes inversely proportional to the carrier density at high-density case, leading to highly efficient nonlinear optical emission from the Si/SiO₂ cuboid. On the other hand, the Si/SiO₂ cuboid could be heated to a high temperature due to the thermalization of hot electrons. A sufficiently high temperature (e.g., ~1500 K) will trigger the intrinsic excitation of carriers in Si, which supplies a huge number of electrons from the valence band to the Δ point of the conduction band[22]. In this case, a significantly enhanced nonlinear optical emission is expected, which is manifested in the burst of hot electron luminescence. It should be emphasized that the key point of this scenario is the generation

of high-density carriers, which is greatly enhanced by exploiting the quasi-BIC supported by the Si/SiO₂ cuboid, as demonstrated in this work.

**Identifying the quasi-BIC of Si/SiO₂ cuboid.** In order to design a Si/SiO₂ cuboid supporting a quasi-BIC, we calculated the scattering spectra for Si/SiO₂ cuboids with variable length ($l$) (Fig. 2a). A TE-polarized wave was used as the excitation source. The width ($w$) and height ($h$) of Si/SiO₂ cuboids were chosen to be 300 and 230 nm, respectively. For all Si/SiO₂ cuboids, the thickness of the outer SiO₂ layer was set to be anisotropic. One can easily identify the ED and MD modes with broad linewidths and invariant resonant wavelengths and the MO (or Fabry-Perot, F-P) mode with a narrow linewidth and a linearly increased resonant wavelength (Supplementary Note 3). We examined the influence of a SiO₂ layer on the scattering spectrum of a Si/SiO₂ cuboid and found that the existence of a SiO₂ layer affects only the wavelength of the quasi-BIC (Supplementary Note 4). By considering the anisotropic oxidation of Si/SiO₂ cuboids during the fabrication process, we can obtain a scattering spectrum in which the quasi-BIC agrees well with that observed in the experiment (Fig. 2b). A remarkable feature is the interference of the ED, MD and MO modes occurring in the wavelength range of 600−800 nm (Supplementary Note 3). As a result, an asymmetric Fano lineshape, which is characterized by an asymmetry parameter $q$[39–41], is observed in the scattering spectrum of each Si/SiO₂ cuboid. It was found that $q$ values are negative for Si/SiO₂ cuboids with $l < 596$ nm and they become positive for $l > 596$ nm. For the Si/SiO₂ cuboid with $l = 596$ nm, the $q$ value approaches positive infinity and a symmetric Lorentz lineshape peaking at ~693 nm is observed, implying the formation of a quasi-BIC (Fig. 2b). We also examined the dependence of the Q factor on the structure parameter of the Si/SiO₂ cuboid and observed the largest Q factor at the quasi-BIC (Supplementary Notes 5.1 and 5.2). Based on the multipolar decomposition[42] of the scattering spectrum, it was revealed that the quasi-BIC is indeed a mixed

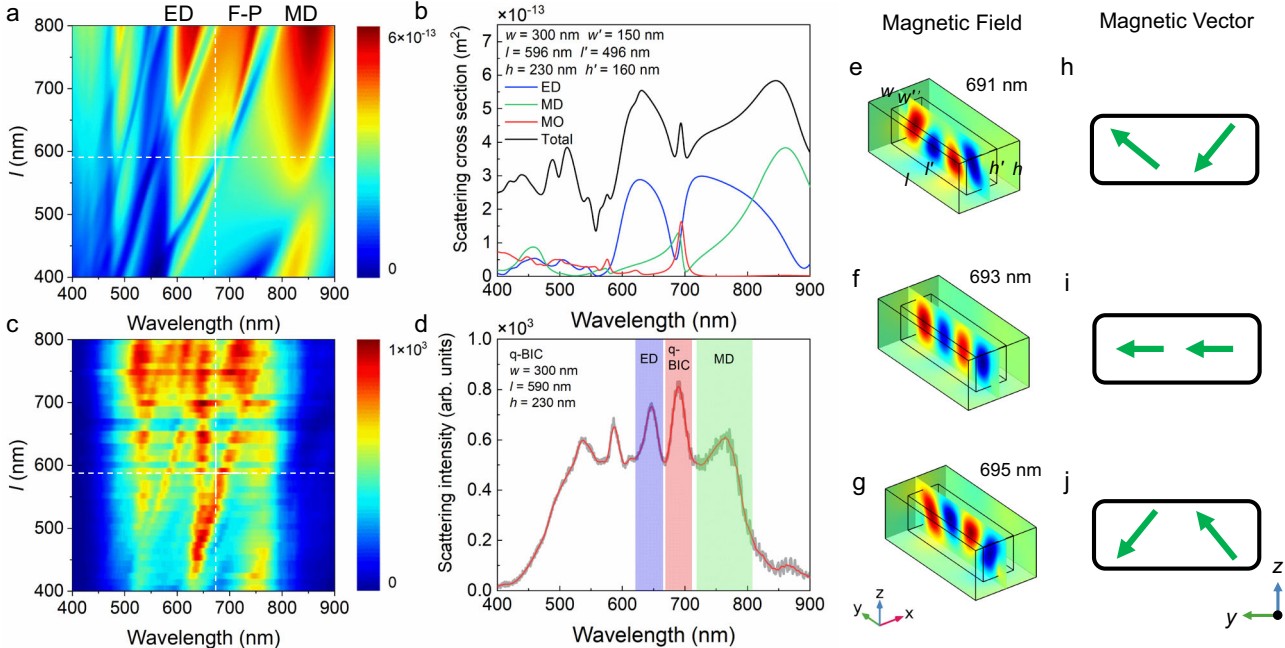

**Fig. 2 Scattering properties of Si/SiO₂ cuboids and the magnetic field distribution at the quasi-BIC. a** Scattering spectra calculated for Si/SiO₂ cuboids with variable length $l$ and fixed width $w = 300$ nm and height $h = 230$ nm, the thicknesses of the outer SiO₂ layer in the x, y, z directions are chosen to be $t_x = 75$ nm, $t_y = 50$ nm, and $t_z = 70$ nm. **b** Scattering spectrum calculated for the Si/SiO₂ cuboid with $l = 596$ nm, $w = 300$ nm and $h = 230$ nm, which has been decomposed into the contributions of the Mie resonances of different orders. **c** Scattering spectra measured for Si/SiO₂ cuboids with the same width of $w \sim 300$ nm and different lengths. The length of the Si/SiO₂ cuboid supporting the quasi-BIC is indicated by the dashed line. **d** Scattering spectrum measured for a Si/SiO₂ cuboid with $l \sim 590$ nm and $w \sim 300$ nm. The scattering peaks corresponding to the ED, quasi-BIC, and MD resonances are marked by colored regions. **e–g**, Magnetic field distributions calculated for the Si/SiO₂ cuboid with $l = 596$ nm and $w = 300$ nm at different wavelengths of 691, 693, and 695 nm. The magnetic vectors corresponding to the magnetic field distributions are shown in **h–j**. Here, ED, F-P, MD, MO represent electric dipole, Fabry-Perot mode, magnetic dipole, and magnetic octupole, respectively.

state of the ED, MD, and MO modes. In addition, it was noticed that the amplitudes of the ED and MD modes are equal at the quasi-BIC. More interestingly, it was found that the phase difference between the ED/MD mode and the MO mode is equal to zero, implying that the three optical modes involved in the interference are in-phase (Supplementary Note 5.3). We measured the scattering spectra for Si/SiO₂ cuboids with different lengths and presented them in the form of two-dimensional scattering intensity (Fig. 2c). The optical resonances observed in the measured scattering spectra are in qualitatively good agreement with those predicted in the simulated ones. The redshift of the MO mode in between the fixed ED and MD modes is clearly identified. The quasi-BIC emerges at the point ($l \sim 590$ nm and $\lambda \sim 690$ nm), where the scattering spectrum evolves from an asymmetric Fano lineshape to a symmetric Lorentz lineshape (Supplementary Note 5.2). In the scattering spectrum of the Si/SiO₂ cuboid with $l = 590$ nm (Fig. 2d), the quasi-BIC is identified as a scattering spike with a symmetric Lorentz lineshape in between the ED and MD resonances. The wavelength of the quasi-BIC observed in the measured scattering spectrum ($\lambda \sim 690$ nm) agrees well with that predicted in the simulated one ($\lambda \sim 693$ nm) based on the anisotropic oxidation model (Fig. 2b). In order to gain a deep insight into the quasi-BIC, we compared the magnetic field distribution in the Si/SiO₂ cuboid at the quasi-BIC with those at the nearby wavelengths (Fig. 2e–g). From the magnetic vectors extracted from the magnetic field distributions (Fig. 2h–j), it was found that the quasi-BIC exhibits a pure MO mode, which is modified significantly when the wavelength deviates slightly from the quasi-BIC. Apart from the inspection of the lineshape evolution in the scattering spectra, the quasi-BIC can also be found by

examining the evolution of the eigenmode or the scattering efficiency (Supplementary Notes 5.4 and 5.5).

**Enhanced nonlinear optical absorption via quasi-BIC.** Basically, the quasi-BICs of Si/SiO₂ cuboids can be employed to enhance the absorption of the laser light or the emission of the luminescence. However, it has been demonstrated that BICs could be easily destroyed by optical doping[43–45]. In our case, the refractive index of Si will be changed significantly by injecting high-density carriers, which eventually leads to the quenching of the quasi-BICs (Supplementary Notes 6 and 7). Fortunately, the quasi-BIC of a Si/SiO₂ cuboid can be used to dramatically enhance the nonlinear optical absorption, making it possible to inject dense electron-hole plasma into the Si/SiO₂ cuboid via a TPA process by using femtosecond laser pulses. Since the injection of carriers can be completed in a short time (~100 fs), the quenching of the quasi-BIC following the thermalization of the injected carriers has no influence on the injection process.

Previously, a physical quantity $I(\lambda) = [\int |E(\lambda)|^4 dV]/V$ is generally used to characterize the TPA of a nanomaterial with a volume of $V$[46]. In order to verify that the largest TPA is achieved at the quasi-BIC of a Si/SiO₂ cuboid, we calculated the wavelength dependent TPA (i.e., $I(\lambda)$) for Si/SiO₂ cuboids with different lengths (Fig. 3a). The evolutions of the ED, MD, and MO modes with increasing length of the Si/SiO₂ cuboid, which are extracted from the decomposition of the scattering spectra, are also presented for reference. It was noticed that the evolution of the maximum TPA in each Si/SiO₂ cuboid follows exactly the evolution of the MO mode. The largest TPA is observed at the quasi-BIC of the Si/SiO₂ cuboid with $l = 740$ nm (marked

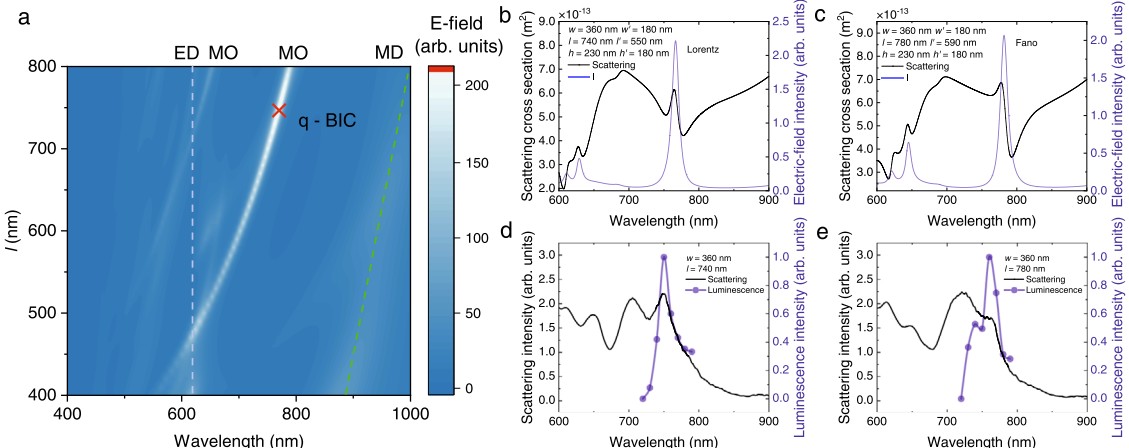

**Fig. 3 Determination of the quasi-BIC. a** Two-photon-induced absorption (TPA) spectra ($\frac{1}{V}\int|E|^4\mathrm{d}V$ spectra) calculated for Si/SiO$_2$ cuboids with $w = 360$ nm and different lengths. ED, MD, MO represent electric dipole, magnetic dipole, and magnetic octupole, respectively. The TPA and scattering spectra calculated for Si/SiO$_2$ cuboids with $l = 740$ and 780 nm are shown in **b**, **c**. The excitation spectra measured for Si/SiO$_2$ cuboids with $l = 740$ and 780 nm are shown in **d**, **e**, respectively. In each case, the scattering spectrum is also provided for reference.

with a red cross). The evolution of the scattering spectrum was also examined experimentally (Supplementary Notes 8). At the quasi-BIC, the interference between the ED, MD and MO modes leads to the strongest electric field intensity inside the Si/SiO$_2$ cuboid, leading to the largest TPA. This behavior is confirmed by inspecting the TPA and scattering spectra of the Si/SiO$_2$ cuboid supporting the quasi-BIC (Fig. 3b). The largest TPA is indeed obtained at the quasi-BIC. Similarly, the maximum TPA is also observed at the quasi-BIC of a Si/SiO$_2$ cuboid, implying the influence of the outer SiO$_2$ layer on the TAP is negligible due to the low refractive index of SiO$_2$ (Supplementary Note 4.3). In this case, it was noticed that the TPA spectrum matches well with the spectrum of a 100-fs femtosecond laser pulse, implying a highly efficient injection of carriers. For comparison, we also calculated the TPA and scattering spectra for a Si/SiO$_2$ cuboid with $l = 780$ nm (Fig. 3c). In this case, the scattering spectrum appears as an asymmetric Fano lineshape. In addition, the scattering peak does not match the peak of TPA, which is reduced by a factor of ~6.0. To confirm the simulation results, we measured the scattering and excitation spectra for two Si/SiO$_2$ cuboids with structure parameters similar to those analyzed above (Fig. 3d, e). The discrepancies between the simulation results and the experimental observations are observed mainly in the scattering spectra and they are caused by two reasons. One is the influence of the SiO$_2$ layers on the surfaces of Si/SiO$_2$ cuboids and the other is the lower quantum efficiency of the detector at long wavelengths (>850 nm). For the Si/SiO$_2$ cuboid with $l = 740$ nm, it was found that the largest TPA is indeed achieved at the quasi-BIC, which appears as a small spike in the scattering spectrum. In comparison, a slight redshift of the TPA peak with respect to that of the scattering peak was observed in the Si/SiO$_2$ cuboid with $l = 780$ nm, in good agreement with the simulation result.

**Lighting up Si/SiO$_2$ cuboids with quasi-BICs**. Since the bandgap energy of Si at the Γ point is ~3.4 eV, the electrons in the valence band can be activated vertically to the conduction band (Γ point) via a TPA process by using femtosecond laser pulses with wavelengths shorter than 730 nm (~1.70 eV). In our experiments, we chose to excite Si/SiO$_2$ cuboids resonantly at their quasi-BICs (~720 nm) by using femtosecond laser pulses with TE polarization (Fig. 4a). We first examined a regular array of Si/SiO$_2$ cuboids under the microscope by using a charge-coupled device

(CCD) (Fig. 4b). The Si/SiO$_2$ cuboid ($l = 440$ nm and $w = 260$ nm) located at the center of the image was excited by using femtosecond laser pulses with a pulse energy of $E = 0.82$ pJ, emitting hot electron luminescence (Fig. 4c). In this case, the Si/SiO$_2$ cuboid appeared as a bright spot in the image. Surprisingly, we observed the burst of luminescence when the excitation pulse energy was raised to $E_{\text{th}} = 0.90$ pJ. In this case, the ultra-bright white light emitted from the Si/SiO$_2$ cuboid exhibited a cross section as large as $20 \times 20$ μm$^2$ (Fig. 4d) (Supplementary Note 9.1 and Supplementary Movies 1 and 2). To gain a deep insight into the luminescence burst phenomenon, we increased the pulse energy of the excitation laser light and examined the change of the luminescence spectrum (Fig. 4e). The luminescence spectra below and above the threshold appear as broadband emissions with enhancements observed at the Mie resonances of the Si/SiO$_2$ cuboid (Supplementary Note 9.2). A significant increase in the luminescence intensity is observed above the threshold. The luminescence burst was clearly reflected in the abrupt increase of the luminescence intensity at a critical excitation pulse energy (Fig. 4f). It should be emphasized that the white light emission from Si nanoparticles, including Si/SiO$_2$ cuboids studied in this work, originates from the interband transition of hot electrons, rather than other physical origins such as electrical discharge. Previously, the enhanced hot electron luminescence from a Si nanoparticle was observed at the MQ/EQ resonances or MD resonance of the Si nanoparticle[18,20]. In addition, the dependence of the luminescence intensity on the excitation irradiance exhibits a slope in between 2.0 and 3.0, verifying the 2PA/3PA process involved in the luminescence[18]. Moreover, the scattering spectra of Si/SiO$_2$ cuboids remain unchanged before and after the luminescence burst, implying no change in the crystalline structure (Supplementary Note 10). Finally, the hot electron luminescence from Si/SiO$_2$ cuboids was not observed when the high-Q quasi-BICs were resonantly excited by using picosecond laser pulses[29]. All these experimental observations indicate undoubtedly that the luminescence from Si/SiO$_2$ cuboids belongs to nonlinear optical emission originating from the interband transition of hot electrons. Based on the previous study, the intrinsic excitation of carriers in the Si/SiO$_2$ cuboid, which is induced by the high temperature resulting from the injection of high-density carriers, is responsible for the exponential growth of the luminescence intensity[22].

In general, the quantum efficiencies of Si nanocrystals are larger than those of porous Si. However, the quantum efficiency

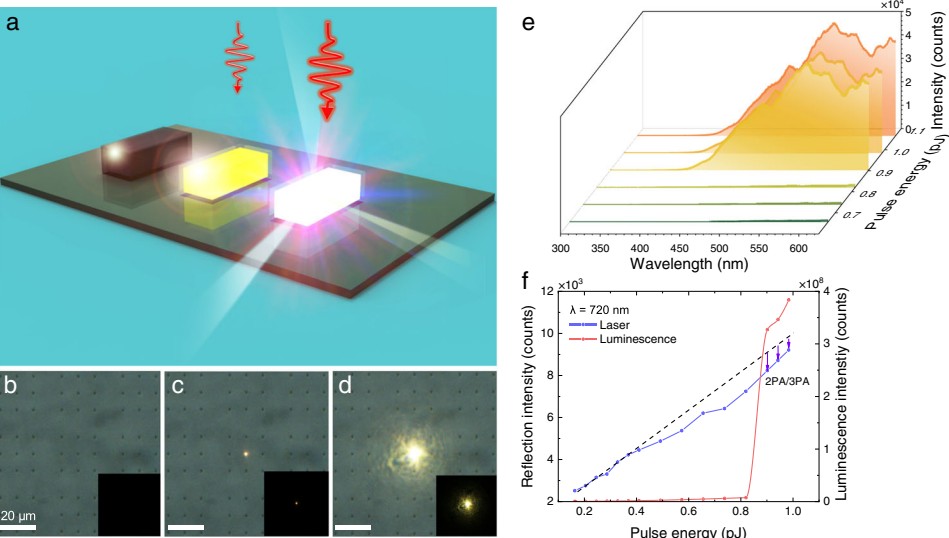

**Fig. 4 Luminescence burst observed for Si/SiO₂ cuboids. a** Schematic showing the burst of hot electron luminescence from a Si/SiO₂ cuboid with increasing excitation pulse energy. **b–d** Evolution of the luminescence with increasing excitation pulse energy observed for a Si/SiO₂ cuboid ($l = 440$ nm, $w = 260$ nm) recorded by using a coupled charge device. **e** Evolution of the luminescence spectrum of the Si/SiO₂ cuboid with increasing pulse energy. **f** Dependences of the luminescence intensity and the reflected laser light intensity on the excitation pulse energy obtained for the Si/SiO₂ cuboid.

values for Si nanocrystals scattered in a very wide range from ~10 to ~100%, depending strongly on the fabrication technique and the characterization method (see Supplementary Note 11). In previous studies, the quantum efficiency was generally evaluated by seeking the ratio of the radiative recombination time to the nonradiative one[47]. In these cases, the modification in the radiative recombination process induced by the Purcell effect or the local density of states need to be taken into account in order to obtain accurate values of quantum efficiency. In our case, we estimated the quantum efficiency of Si/SiO₂ cuboids by measuring the numbers of the emitted and absorbed photons. The absorption of Si/SiO₂ cuboids, which is dominated by nonlinear optical absorption at high excitation densities, offers us the opportunity to accurately extract the number of absorbed photons (see Supplementary Note 12). Thus, how to accurately estimate the number of emitted photons becomes a key point because only a fraction of emitted photons from a Si/SiO₂ cuboid was detected. In this work, we simulated the collection efficiency of the emitted photons by considering both the directivity of the luminescence and the numerical aperture of the objective. An average collection efficiency of ~58% was obtained by using this method. We also inspected the dependence of the excitation laser light reflected from the substrate (with the Si/SiO₂ cuboid) on the pulse energy (Fig. 4f). It is noticed that the optical absorption of the Si/SiO₂ cuboid, which is governed by linear absorption at low pulse energies, will become dominated by nonlinear absorption at high pulse energies. This unique feature offers us the opportunity to extract the number of absorbed photons from the deviation of the reflection intensity from the linear relationship observed at low pulse energies. The slope of this linear relationship can be calibrated by measuring the reflection intensities from the substrate only at different pulse energies. In this way, the external quantum efficiency for the luminescence of the Si/SiO₂ cuboid, which is defined as the ratio of the number of photons emitted out of the Si/SiO₂ cuboid to the number of absorbed photons, is found to be ~13% (Supplementary Note 12). This value is further improved when comparing with the previous results for Si nanoparticles on an Ag film[22]. The internal quantum efficiency, which is given by the ratio of the number of photons generated

inside the Si/SiO₂ cuboid to the number of absorbed photons, should be larger than the external quantum efficiency.

Although an improved quantum efficiency was achieved in Si/SiO₂ cuboids by utilizing quasi-BICs, this value is still much smaller than those observed for GaAs low-dimensional materials, including quantum wells, superlattices, nanowires, and quantum dots[48–51]. Recently, the hot electron luminescence from GaAs nanoparticles that support Mie resonances was also investigated[52]. It was found, however, that the hot electron luminescence from GaAs nanoparticles excited by femtosecond laser pulses originates from the intraband transition of hot electrons, similar to the hot electron luminescence from Au or Ag nanoparticles[53]. In contrast, the hot electron luminescence from Si nanoparticles arises from the interband transition of hot electrons[18]. Owing to the different bandgap energies and band structures of GaAs and Si, it is difficult to make a fair comparison between the quantum efficiencies of GaAs and Si nanoparticles. In addition, the physical mechanisms for the radiative recombination are also different in GaAs and Si nanoparticles. More importantly, it is difficult to obtain high-quality GaAs nanoparticles with distinct Mie resonances even though a post annealing process is employed. Very recently, it was noticed that lasing from GaAs nanodisks was realized by exploiting the quasi-BICs[37,54,55]. Although the quantum efficiency of GaAs nanodisks was not investigated in this case, the realization of lasing action implies that the quantum efficiency of GaAs nanoparticles should be higher than that of Si nanoparticles.

We examined the polarization of the luminescence emitted from Si/SiO₂ cuboids by inserting a polarization analyzer in the collection channel. It was found that the luminescence of the Si/SiO₂ cuboid exhibits a linear polarization perpendicular to the length of the Si/SiO₂ cuboid. We also simulated the three-dimensional radiation pattern of a Si/SiO₂ cuboid and found that the emission from the Si/SiO₂ cuboid is governed by the radiations from ED and MQ (Supplementary Note 13).

Basically, the transient absorption spectra for Si/SiO₂ cuboids can be achieved by using the so-called pump-probe technique. In this case, a supercontinuum with a broadband covering the visible to near infrared spectral range, which is commonly generated by femtosecond laser pulses, is necessary. Since the optical

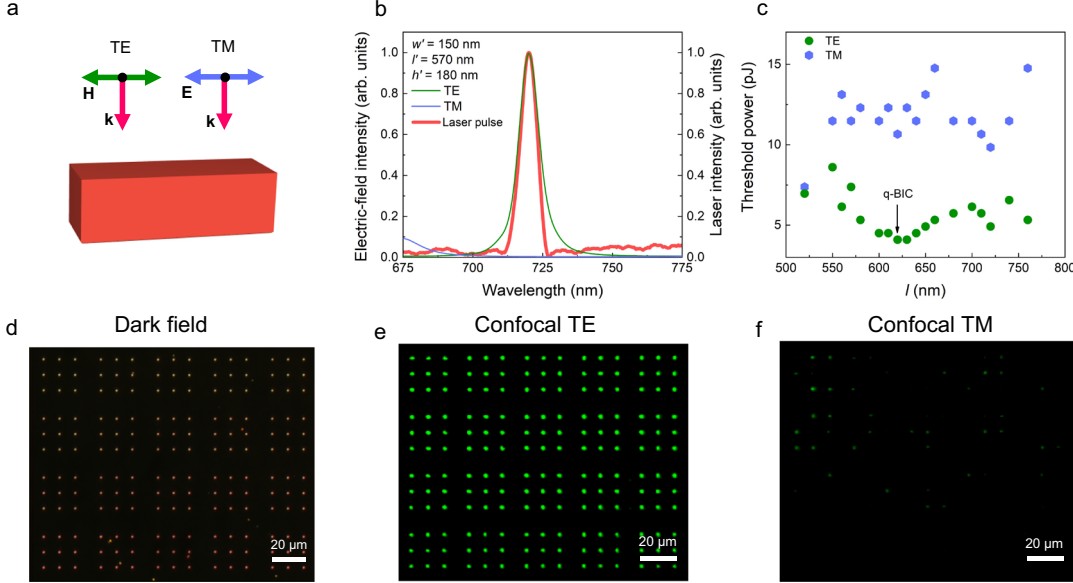

**Fig. 5 Polarization dependent threshold for the luminescence burst. a** Schematic illustrating the excitation of a Si/SiO$_2$ cuboid with transverse electric (TE) and transverse magnetic (TM) polarized laser light. **b** Two-photon-induced absorption (TPA) ($\frac{1}{V}\int |E|^4 dV$) spectrum calculated for a Si/SiO$_2$ cuboid with $l' = 570$ nm and $w' = 150$ nm. Also shown is the spectrum of a femtosecond laser pulse with a duration of 130 fs at 720 nm. **c** Thresholds for the luminescence burst measured Si/SiO$_2$ cuboids with different lengths excited by using TE- and TM-polarized laser light. **d** Dark-field image of a regular array of Si/SiO$_2$ cuboids. **e** Luminescence from the regular array of Si/SiO$_2$ cuboids shown in **d** observed by using a laser scanning confocal microscope with TE-polarized laser light at 720 nm as the excitation source. The luminescence observed by using TM-polarized laser light is shown in **f**.

characterizations of Si/SiO$_2$ cuboids are performed under a microscope, such a pump-probe measurement remains a big challenge at present. As an alternative, we calculated the absorption spectra of a Si/SiO$_2$ cuboid at different injected carrier densities. The quenching of the quasi-BIC was observed at high carrier densities (see Supplementary Note 6). We also measured the luminescence lifetimes of Si/SiO$_2$ cuboids. It was found that the luminescence lifetime, which is ~110 ps below the threshold, is reduced to be ~49 ps after the luminescence burst (see Supplementary Note 14).

**Dependence of excitation efficiency on laser polarization**. It should be emphasized that the optical resonances supported by Si/SiO$_2$ cuboids, such as the Mie resonances and the quasi-BICs studied in this work, play a key role in dramatically improving the quantum efficiency of Si/SiO$_2$ cuboids. Basically, the optical modes excited in a Si/SiO$_2$ cuboid are quite sensitive to the polarization of the excitation laser light (Fig. 5a). Here, we examined a Si/SiO$_2$ cuboid with the quasi-BIC at ~720 nm, which is suitable for injecting dense electron-hole plasma (Supplementary Note 15). We calculated the TPA spectra for a Si/SiO$_2$ cuboid with $l' = 570$ nm (Fig. 5b), which is excited by using TE- and TM-polarized light, respectively. It can be seen that a quasi-BIC appears at ~720 nm for the TE-polarized light. In addition, its spectrum matches well with that of the femtosecond laser pulses used to excite the Si/SiO$_2$ cuboid in the experiments. In sharp contrast, the TPA for the TM-polarized light is much smaller at this wavelength. We performed the statistics for the luminescence burst thresholds measured for Si/SiO$_2$ cuboids with different lengths excited by using TE- and TM-polarized light at 720 nm (Fig. 5c). It was noticed that the threshold for the TE excitation is smaller by a factor of 2.0–3.0 as compared with that for the TM excitation. In addition, it was found that the lowest threshold for luminescence burst is achieved in Si/SO$_2$ cuboids with $l \sim 620$ nm, which support quasi-BICs. These experimental observations are in good agreement with the theoretical analysis results based on numerical simulation. To further verify the polarization dependent excitation efficiency of Si/SiO$_2$ cuboids, we examined the luminescence of periodically arranged Si/SiO$_2$ cuboids with a confocal laser scanning microscope. The regular array of Si/SiO$_2$ cuboids was examined firstly by using a dark-field microscope (Fig. 5d). We obtained the luminescence images of the Si/SiO$_2$ cuboids excited by using TE- and TM-polarized light (Fig. 5e, f). For TE-polarized light, all the Si/SiO$_2$ cuboids in the array were lightened up at an excitation pulse energy of $E = 4.1$ pJ. In sharp contrast, only a small part of Si/SiO$_2$ cuboids in the array emitted weak luminescence when TM-polarized was employed. This result indicates clearly that TE-polarized light is more efficient for lighting up Si/SiO$_2$ cuboids (Supplementary Note 16).

**Discussion**

In summary, we proposed the use of the quasi-BICs of single Si/SiO$_2$ cuboids for injecting dense electron-hole plasma into Si/SiO$_2$ cuboids via a TPA process, which in turn trigger the intrinsic excitation of carriers and significantly improve the quantum efficiency of Si/SiO$_2$ cuboids. We examined the optical modes supported by Si/SiO$_2$ cuboids with different geometrical parameters and determined the quasi-BIC as a mixed state of the ED, MD and MO modes with zero phase difference. We observed ultra-bright hot electron luminescence coming from the Si/SiO$_2$ cuboid when the excitation pulse energy exceeds a threshold and obtained a quantum efficiency as high as ~13%. We found that TE-polarized is more efficient for generating hot electron luminescence in Si/SiO$_2$ cuboids. Our findings pave the way for fabricating on-chip white light sources for photonic integrated circuits in the future and open new horizons for manipulating the luminescence properties of semiconductors with indirect bandgaps.

**Methods**

**Sample fabrication**. We fabricated Si/SiO$_2$ cuboids from a Si-on-sapphire wafer (SOS), which is formed by a 230-nm-thick c-Si and a 500-μm-thick sapphire, accorrding to the following procedure. (see Supplementary Note 1): (1) A negative resist (hydrogen silsesquioxane, HSQ) with a thickness of 250 nm was spin-coated

on the SOS wafer; (2) An array of Si/SiO$_2$ cuboids was defined in HSQ by using electron-beam lithography (Vistec EBPG-5000plusES, Raith) at 100 keV; (3) After developing in tetramethylammonium hydroxide, the patterns were transferred to the Si by using inductively coupled plasma etching (PlasmaPro System 100ICP180, Oxford Instruments); (4) The remaining HSQ was removed by hydrogen fluoride acid; (5) In order to eliminate the surface defects introduced in Si cuboids in the etching process, a thermal oxidation process was performed at 1000 °C for 100 min, forming a SiO$_2$ layer (~60–80 nm) on the surface of each Si/SiO$_2$ cuboid.

In the fabrication process, Si/SiO$_2$ cuboids with the same geometrical parameters (length and width) were arranged as a $3 \times 3$ array. The distance between the two neighboring Si/SiO$_2$ cuboids was designed to be $d = 10$ μm so that each Si/SiO$_2$ cuboid is isolated from its neighbors. The morphologies of the fabricated Si/SiO$_2$ cuboids were examined by using SEM observations (see Supplementary Note 2).

**Optical characterization**. The linear and nonlinear optical responses of Si/SiO$_2$ cuboids with different geometrical parameters were characterized by using an inverted microscope (Observer A1, Zeiss) equipped with white light and femtosecond laser light as excitation sources.

A dark-field microscope with a home-built oblique incidence system was employed to characterize the scattering properties of Si/SiO$_2$ cuboids. In this case, the illumination light was incident on Si/SiO$_2$ cuboids at an angle of ~40° and the forward scattering light was collected by the objective of the dark-field microscope (see Supplementary Note 17).

The Si/SiO$_2$ cuboids were excited by the femtosecnd laser light focused with the 100× objective of the microscope. The generated luminescence was gathered by the objective and directed to a spectrometer (SR-500i-B1, Andor) for spectral analysis or to a charge-coupled device (DU970N, Andor) for imaging. The mapping of the hot electron luminescence from an array of Si/SiO$_2$ cuboids was performed by using a confocal laser scanning microscope (A1MP, Nikon).

**Numerical calculation**. The scattering spectra of Si and Si/SiO$_2$ cuboids and the corresponding electric and magnetic field distributions were calculated based on the finite element method (Multiphysics, COMSOL) and the finite-difference time-domain method (FDTD solution, Lumerical). Although the Maxwell equations were solved in frequency and time domains, respectively, very good agreements were found between the simulation results obtained by using these two methods. By using the finite element method, we could easily derive the integration of the electric field over a Si/SiO$_2$ cuboid (e.g., $\int |E(\lambda)|^4 dV / V$), which characterizes the nonlinear optical absorption of the Si/SiO$_2$ cuboid. On the other hand, the decay of electric field inside a Si/SiO$_2$ cuboid, which gives the Q factor of the Si/SiO$_2$ cuboid, could be readily obtained by using the FDTD method.

In the numerical simulations, the height of Si cuboids was fixed to be $h = 180$ nm while the length and width of Si cuboids were varied in order to find out the quasi-BIC suitable for the excitation of Si/SiO$_2$ cuboids. This height corresponds to the thickness of the Si layer in the SOS wafer used for the fabrication of Si/SiO$_2$ cuboids after the thermal oxidation (see more details in Supplementary Note 1). The refractive index of Si was taken from Aspnes[56] while those of SiO$_2$ and Al$_2$O$_3$ were chosen to be 1.45 and 1.70. In each case, the Si/SiO$_2$ cuboid located on an Al$_2$O$_3$ substrate was placed at the center of the simulation region, which was enclosed by a perfectly matched layer (PML) that absorbs completely the outgoing light. The refractive index of the surrounding medium (air) was set to be 1.00. The dimensions of the air layer and the Al$_2$O$_3$ substrate in the simulation region were made to be larger than the three times of the incident light wavelength. When we used the COMSOL Multiphysics for numerical simulation, free tetrahedral meshes were employed in the simulation region while cuboid meshes were used in the perfectly matched layer. In comparison, we used Yee grid in the FDTD simulations. In order to obtain accurate results, the maximum mesh size was set to be 1.0 nm in both cases. The electric and magnetic field distributions in the Si/SiO$_2$ cuboid were extracted from the field detectors inserted in it. The Q factor of the Si/SiO$_2$ cuboid was extracted by monitoring the field decay inside the Si/SiO$_2$ cuboid or by fitting the scattering spectrum of the Si/SiO$_2$ cuboid (see Supplementary Note 5.1 and 5.2).

The multipolar expansion method[42] was employed to decompose the total scattering of a Si/SiO$_2$ cuboid into the contributions of electric and magnetic modes of different orders, including ED, MD, EQ, MQ, EO, and MO etc.

**Reporting summary**. Further information on research design is available in the Nature Research Reporting Summary linked to this article.

## Data availability
The data that support the findings of this study are available from the corresponding authors upon reasonable request.

## Code availability
The codes and simulation files that support the plots and data analysis within this paper are available from the corresponding author upon reasonable request.

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

## Acknowledgements

S. Lan acknowledges the financial support from the National Natural Science Foundation of China (Grant Nos. 11874020 and 12174123). J.L. acknowledges the financial support by the National Natural Science Foundation of China (Grant No. 11974436) and the Guangdong Basic and Applied Basic Research Foundation (No. 2020B1515020019). J.S. acknowledges the financial support by the National Natural Science Foundation of China (Grant No. 61874144) and the Guangdong Basic and Applied Basic Research Foundation (No. 2018B030311045).

## Author contributions

S. Lan, J.L., M.P., and J.X. conceived the idea. J.L, J.S., Y.R., and M.Z. fabricated the samples. M.P., S. Li, and X.H. carried out the optical experiments. M.P. and J.X. performed the numerical modeling. S. Lan, M.P., J.X., and J.L. analyzed the data and wrote the manuscript. S. Lan and J.L. supervised the project. All the authors read and commented on the manuscript.

## Competing interests

The authors declare no competing interests.
