## [Peer review file · Nature Communications]

REVIEWER COMMENTS

Reviewer #1 (Remarks to the Author):

I read the paper “Highly efficient nonlinear optical emission from a subwavelength crystalline silicon cuboid mediated by supercavity mode” by Mingcheng Panmai et al with great interest. The Authors analyzed the photoluminescence efficiency in silicon cubits under femtosecond laser pulse excitation. They achieve around 14% efficiency for the silicon cuboid supporting quasi-bound states in the continuum also known as supercavity modes. The paper looks solid and novel. All the details of the simulation, fabrication, and experiment are comprehensively described in the supporting information. Nevertheless, I have several questions and comments impeding to recommend this paper for publication in Nat. Comm.

I would like to encourage the Authors to address the following questions and comments:

1. It is not clear from the abstract what do the Authors imply under “transverse electric” and “transverse magnetic polarization”. [1] [SEP]
2. I think that it is reasonable to mention that Ge QD in Si can be also used as a light source at the telecommunication range, and the photoluminescence efficiency can be enhanced using the photonic structures supporting bound states in the continuum [Laser & Photonics Reviews, 2000242 (2021); ACS Nano 11, 10704–10711 (2017)]. [1] [SEP]
3. Nanoparticles made of Si/Au alloys with the laser ablation can be also used as a source of white light [Nano letters 18 (1), 535-539 (2018)]. [1] [SEP]
4. Could the Authors compare the achieved quantum efficiency with GaAs? [1] [SEP]
5. In the introduction part, the Authors state that “radiative recombination is mediated by the electric and magnetic quadrupole resonances” without any references or clarifications. Could the Authors comment on this point in more detail? [1] [SEP]
6. In addition to Fano parameter analysis, I would like to ask the Authors to plot the dependence of the Q-factor on the aspect ratio of the cuboid similarly to that was done in [Science 367 (6475), 288-292 (2020); Adv. Materials 33, 2003804 (2021)]. This will be the direct proof that the Authors deal with the quasi-BIC for which the Q factor should be maximal. This should be done both experimentally and theoretically in my opinion. This is the key question. [1] [SEP]
7. BIC (bound state in the continuum is a mode), thus, the expression “BIC mode” sounds not completely correct. [1] [SEP]

8. I don't think that the term "the collapse of the Fano resonance" is used properly. Usually, this term implies that the resonance becomes infinitely narrow and the quasi-BIC turns into the genuine BIC but this is not the case considered in the paper.

9. What the Author can say about the polarization and directivity pattern of the generated light?

Reviewer #2 (Remarks to the Author):

Panmai and coauthors present a joint experimental and theoretical study of Si/SiO₂ cuboids, exploring the properties of a longitudinal quasi-BIC mode in these semiconductor cavities for the purpose of strongly enhancing the photoemission of Si. Given the sensitivity of the quasi-BIC mode to optical excitation, they focus on the response under nonlinear two-photon excitation achieved with ultrashort pulses. The experimental setup clearly shows a strong photoemission under high-power excitation targeting this mode, with properties supported by the theoretical data. The authors report a quantum efficiency above 13%.

The manuscript and its complementing SI are very thorough in reporting the response of the system, characterizing its behaviour through theoretical means, and discussing several divergences between some theoretical models and experiment. These results are, in my opinion, useful to continue the study and exploitation of Si systems in the development of future technology, and this manuscript should be of interests to many groups. Thus, conditional on the authors addressing some specific issues, I would recommend this manuscript for publication in Nature Communications.

1) The quantum efficiency of the material is only compared with another system made by some of the same authors, using Si deposited over a Ag film and a thin SnO₂ spacer. Although this comparison is reasonable, specially given that in both cases they used a similar method to estimate the quantum efficiency, and eminently useful, a broader overview of the literature would be expected to contextualize this value. This should also include systems with nominally larger quantum efficiencies, such as [DOI: 10.1103/PhysRevB.73.132302]. Then, it would be useful to actively discuss the important features and differences between this work and others. As it stands, this work feels largely decontextualized.

2) The method for quantifying the quantum efficiency depends strongly on the linear fit shown in Fig. 5f and S13, so the reported quantum efficiency is sensitive to the specific details and assumptions behind this fit, and they should be discussed in more detail. For instance, in selecting the initial linear trend, what constitute "low pulse energies"? Theoretically, what is the expected behavior of the reflection with respect to pulse energy in the presence of the cuboids? Would it be possible, and useful, to obtain such data without the cuboids and use it as the baseline, instead of the linear fit?

3) Overall, I think that the methodology is not reported in sufficient detail. This is so for the experiment, but most especially for the theory.

4) I believe that the similarity between Fig. 3b,c (theory) and Fig. 3d,e (experiment) is significantly overstated. The connection should be further discussed.

5) It could be informative to expand on the qualitative differences between theory and experiment in Fig. 2. Although the anisotropy discussed in the SI does explain the shift in the qBIC, the experimental scattering presents some overall different features to the theory. Would it be useful to recreate Fig. 2b with the anisotropic Si/SiO₂ cuboid?

Reviewer #3 (Remarks to the Author):

In this work, the authors describe Si cuboids as efficient emitters of white light upon two-photon excitation of a hot electron plasma. They show that cuboids with specific dimensions can sustain bound states in the continuum upon pulsed femtosecond optical pumping due to good overlap of the pump pulse with the optical resonances that are sustained in the cuboid. By matching the cuboid size to a resonance that is slightly larger in frequency than half on the Si band gap, a strong hot electron plasma is achieved that leads to white light emission. This excitation is more efficient with TE than TM modes.

The overlap of optical modes in the cuboid with the TPA and the line shapes of the scattering spectra are discussed in great detail and the reasoning is convincing.

Overall, the work is very interesting and timely.

However, for publication in Nature Comm, I would request a deeper discussion of the process underlying the light emission. What are the involved timescales here? How can the profile of the emission spectrum be understood? I suggest to include a scheme that illustrates the excitation and creation of the plasma and the channels for relaxation (bright and dark)

And can an intuitive picture for the emission process be provided? A figure giving a schematic description would be valuable.

Does the emission stem from transitions in between electronic levels of the Si, or does it come from electrical discharge (maybe generated by displacement charges)?

Here, the spectral profile at or just below the threshold (like Fig 4e) would be interesting to see.

Could it be that optically induced displacement charges are involved in the white emission? In the sense that a strong field is build up between the two ends of the cuboid that leads to electrical discharge generating the white light?

Time resolved optical spectroscopy would be very interesting, both transient absorption and the PL decay traces to get deeper insight into the emission process.

Minor comments:

- I have difficulties to follow the transition from the Fano line shape to the Lorentzian on in Figure 4c. Can the authors show more line plots?

- The threshold value for the pulse energy on top of page 9 does not match the threshold in Figure 4e

- On page 3 in the introduction the authors relate increased Auger processes to increase in quantum efficiency . That is very unusual, as far as I know Auger processes are detrimental for radiative recombination. Can the authors elaborate on this?

- English language needs improvements

- Overall, manuscript needs a careful correction for typos and inconsistencies, for example the caption of Fig2 has 3 typos, and a confusion on the value of w . Also in Figure S11 the l and w labels are not clear.

In summary, I can recommend publication after these points are addressed.

Responses to the comments of the reviewer #1

(Manuscript ID: NCOMMS-21-32514-T)

We would like to thank the comments and suggestions of the reviewer which are definitely helpful for improving the quality of the manuscript. The responses (abbreviated as R) to these comments (abbreviated as C) and the changes made in the revised manuscript (marked in red color) are described in the following.

C1: I read the paper “Highly efficient nonlinear optical emission from a subwavelength crystalline silicon cuboid mediated by supercavity mode” by Mingcheng Panmai et al with great interest. The Authors analyzed the photoluminescence efficiency in silicon cubits under femtosecond laser pulse excitation. They achieve around 14% efficiency for the silicon cuboid supporting quasi-bound states in the continuum also known as supercavity modes. The paper looks solid and novel. All the details of the simulation, fabrication, and experiment are comprehensively described in the supporting information. Nevertheless, I have several questions and comments impeding to recommend this paper for publication in Nat. Comm.

R1: We appreciate the positive report of the reviewer.

C2: It is not clear from the abstract what do the Authors imply under “transverse electric” and “transverse magnetic polarization”.

R2: We would like to thank the reviewer for this question. In the revised manuscript, we have indicated clearly what transverse electric and magnetic polarizations are (“**We revealed that femtosecond laser light with transverse electric polarization (i.e., the electric field perpendicular to the length of a Si cuboid) is more efficient for generating hot electron luminescence in Si cuboids as compared with that of transverse magnetic polarization (i.e., the magnetic field perpendicular to the length of a Si cuboid).**”, see the Abstract).

C3: I think that it is reasonable to mention that Ge QD in Si can be also used as a light source at the telecommunication range, and the photoluminescence efficiency can be enhanced using the photonic structures supporting bound states in the continuum [Laser & Photonics Reviews, 2000242 (2021); ACS Nano 11, 10704–10711 (2017)].

R3: We would like to thank the reviewer for this suggestion and we agree with the reviewer on this point. In the revised manuscript, we have added a sentence to mention this point in the introduction part (“In addition, it was shown that the photoluminescence from germanium (Ge) quantum dots embedded in Si matrix, which is employed as the light source at telecommunication wavelength, was enhanced by using photonic structures supporting BICs^{32,33}.”, see 2nd paragraph in page 4). In addition, the two papers mentioned by the reviewer have been added in references (see Refs. 32 and 33).

C4: Nanoparticles made of Si/Au alloys with the laser ablation can be also used as a source of white light [Nano letters 18 (1), 535-539 (2018)].

R4: We agree with the reviewer on this point. According to the suggestion of the reviewer, we have added a sentence to mention this point in the introduction part (“In addition, efficient white light emission was also observed in Au/Si alloy nanoparticles fabricated by using laser ablation¹⁴.”, see 1st paragraph in page 3). In addition, the paper mentioned by the reviewer has been added in references (see Ref. 14).

C5: Could the Authors compare the achieved quantum efficiency with GaAs?

R5: We would like to thank the reviewer for this suggestion. It is well known that GaAs is a semiconductor with direct bandgap while Si is a semiconductor with indirect bandgap. Therefore, the quantum efficiency of bulk Si ($\sim 10^{-7}$) is several orders of magnitude smaller than that of bulk GaAs ($\sim 60\%$, see *Gallium Arsenide Semiconductor Materials* C Chao, 1967). It has been demonstrated that the quantum efficiency of GaAs can be further enhanced in low-dimensional materials, such as GaAs quantum wells, ($\sim 60\%$ [*Appl. Phys. Lett.* **58**, 2264 (1991)]; $\sim 92\%$ [*Appl. Phys. Lett.* **59**, 857 (1991)]), superlattices ($\sim 99.7\%$ internal quantum efficiency and 72% external quantum efficiency [*Appl. Phys. Lett.* **62**, 131 (1993)]; near unity [*Phys. Status Solidi. RRL*, **15**, 2100106 (2021)]), nanowires ($\sim 100\%$ [*Nat. Commun.* **7**, 11927 (2016)]), and quantum dots ($\sim 91.4\%$ internal quantum efficiency [*Appl. Phys. Lett.* **82**, 841 (2003)]). Recently, the hot electron luminescence from GaAs nanoparticles that support Mie resonances was also investigated (see Ref. 52). It was found, however, the hot electron luminescence from GaAs nanoparticles excited by femtosecond laser pulses originates from

the intraband transition of hot electrons, similar to the hot electron luminescence from metallic (Au or Ag) nanoparticles (see Ref. 53). In contrast, the hot electron luminescence from Si nanoparticles arises from the interband transition of hot electrons (see Ref. 18). This difference is reflected in the slope extracted from the luminescence intensity on the excitation irradiance of GaAs or Si nanoparticles. For GaAs nanoparticles, a linear relationship between the slope and the energy of the emitted photons is observed (see Ref. 52). Differently, a constant slope between 2 and 3 is found for Si nanoparticles (see Ref 18). Owing to the different bandgap energies and band structures of GaAs and Si, it is difficult to make a fair comparison between the quantum efficiencies of GaAs and Si nanoparticles. In addition, the physical mechanisms for the radiative recombination are also different in GaAs and Si nanoparticles. More importantly, it is difficult to obtain high-quality GaAs nanoparticles with distinct Mie resonances even though a post annealing process is employed (see Ref. 52). Very recently, it was noticed that lasing from GaAs nanodisks was realized by exploiting the quasi-BICs (see Ref. 54). Although the quantum efficiency of GaAs nanodisks was not investigated in this case, the realization of lasing action implies that that the quantum efficiency of GaAs nanoparticles should be higher than that of Si nanoparticles. To address this comment of the reviewer, we have added a detailed discussion in the revised manuscript (“Although an improved quantum efficiency was achieved in Si cuboids by utilizing quasi-BICs, this value is still much smaller than those observed for GaAs low-dimensional materials, including quantum wells, superlattices, nanowires, and quantum dots⁴⁸⁻⁵¹. Recently, the hot electron luminescence from GaAs nanoparticles that support Mie resonances was also investigated⁵². It was found, however, the hot electron luminescence from GaAs nanoparticles excited by femtosecond laser pulses originates from the intraband transition of hot electrons, similar to the hot electron luminescence from Au or Ag nanoparticles⁵³. In contrast, the hot electron luminescence from Si nanoparticles arises from the interband transition of hot electrons¹⁸. Owing to the different bandgap energies and band structures of GaAs and Si, it is difficult to make a fair comparison between the quantum efficiencies of GaAs and Si nanoparticles. In addition, the physical mechanisms for the radiative recombination are also different in GaAs and Si nanoparticles. More importantly, it is difficult to obtain high-quality GaAs nanoparticles with distinct Mie resonances even though a post annealing process is

employed⁵². Very recently, it was noticed that lasing from GaAs nanodisks was realized by exploiting the quasi-BICs^{37,54,55}. Although the quantum efficiency of GaAs nanodisks was not investigated in this case, the realization of lasing action implies that that the quantum efficiency of GaAs nanoparticles should be higher than that of Si nanoparticles.”, see 3th paragraph in page 9). In addition, the relevant papers have been added in references (see Refs. 37, 48–55).

C6: In the introduction part, the Authors state that “radiative recombination is mediated by the electric and magnetic quadrupole resonances” without any references or clarifications. Could the Authors comment on this point in more detail?

R6: We are sorry for the missing of references for this statement. As shown in Figs. 2 and 3 of Ref. 18 [*Nature Communications* **9**, 2964 (2018)], the hot electron luminescence from a Si nanoparticle is enhanced at the MQ/EQ resonances of the Si nanoparticle due mainly to the Purcell effect. For the convenience of the reviewer, we have reproduced the figures in the following. To address comment, we have added the reference in the revised manuscript (“As a result, the significantly-enhanced Auger recombination process in combination with the accelerated radiative recombination processes mediated by the electric and magnetic quadrupole resonances increase the quantum efficiency by several orders of magnitude¹⁸.”, see 2nd paragraph in page 3 and Ref. 18).

Fig. 2 White-light emission from silicon nanospheres. **a** Scattering spectra measured for a silicon nanosphere (NS) with $d = 192$ nm. The inset shows the scanning electron microscope image of the silicon NS. **b-d** show the dependence of the nonlinear response spectrum of the silicon NS on the excitation pulse energy measured at 800, 775, and 758 nm, respectively. White-light emission recorded by a charge coupled device is shown in the inset of **d**

When the excitation wavelength λ_{ex} was tuned to 775 nm, a significant increase of the up-converted luminescence was observed at the MQ and EQ resonances (Fig. 2c). For resonant

Fig. 3 Emission enhancement and luminescence lifetime. **a** Nonlinear response spectra measured at different excitation pulse energies for a silicon nanosphere (NS) with $d = 210$ nm. The spectrum of I calculated for the silicon NS is also presented for comparison. The inset shows the dependence of the up-converted luminescence on the excitation pulse energy plotted in a double-logarithmic coordinate. **b** Decay of the up-converted luminescence measured for a silicon NS with $d = 190$ nm after the excitation of the femtosecond laser pulses. The luminescence lifetime is derived to be ~ 52 ps based on a reconvolution fitting analysis. Here, IRF represents instrument response function and χ^2 is a parameter characterizing the fitting quality. **c** The corresponding residuals for the fitting

from the conduction band minimum at the Δ point to the valence band maximum). From the enhancement factors of electric field estimated for the EQ/MQ resonances (see Fig. 3a), it is expected that τ_r can only be reduced by about one order of magnitude from 10–100 ns to 1–10 ns. Although it is anticipated that τ_{nr} can be

C7: In addition to Fano parameter analysis, I would like to ask the Authors to plot the dependence of the Q-factor on the aspect ratio of the cuboid similarly to that was done in [Science 367 (6475), 288-292 (2020); Adv. Materials 33, 2003804 (2021)]. This will be the direct proof that the Authors deal with the quasi-BIC for which the Q factor should be maximal. This should be done both experimentally and theoretically in my opinion. This is the key question.

R7: We would like to thank the reviewer for this good suggestion. We have calculated the dependence of the quality (Q) factor on the aspect ratio of a Si cuboid (l/w), as shown in Fig. R1a,b. In the numerical simulations, the SiO₂ layer (75 nm) on the surface of the Si cuboid is taken into account (i.e., a Si/SiO₂ cuboid). For each Si cuboid, the Q factor is derived by two different methods. One is based on the energy decay of a dipole placed in the Si cuboid. The other relies on the fitting of the calculated scattering spectra. The Q factors obtained by using these two methods are quite similar and the largest value is achieved at the quasi-BIC (~600 nm). In Fig. R1c, we present the dependence of Q factor on the aspect ratio derived from the scattering spectra measured for Si cuboids. Although the Q factors are smaller than those derived from the simulation results (due mainly to the imperfections of Si cuboids induced in the fabrication process and the various measurement errors induced in the optical characterization), it is noticed that the maximum Q factor is observed at the quasi-BIC (~600 nm). In the revised manuscript, we have added a brief discussion on this issue (“We also examined the dependence of the quality (Q) factor on the structure parameter of the Si/SiO₂ cuboid and observed the largest Q factor at the quasi-BIC (Supplementary Notes 5.1 and 5.2).”, see 2nd paragraph in page 6). In addition, the dependence of the quality (Q) factor on the aspect ratio of a Si cuboid and a detailed description for the methods used to derive the Q factors have been added in Supplementary Materials (see Supplementary Note 5.1). In addition, the paper mentioned by the reviewer has been added in references (see Refs. 39 and 41).

Fig. R1 (a) Dependence of the Q factor on the length or aspect ratio (l/w) of the Si/SiO₂ cuboid calculated by using (a) numerical simulation based on the FDTD method, (b) fitting of the simulated scattering spectra of Si/SiO₂ cuboids, and (c) fitting of the measured scattering spectra of Si/SiO₂ cuboids.

C8: BIC (bound state in the continuum is a mode), thus, the expression “BIC mode” sounds not completely correct.

R8: We would like to thank the reviewer for this comment. In the revised manuscript, we have replaced “BIC mode” with “BIC”.

C9: I don’t think that the term “the collapse of the Fano resonance” is used properly. Usually, this term implies that the resonance becomes infinitely narrow and the quasi-BIC turns into the genuine BIC but this is not the case considered in the paper.

R9: We would like to thank the reviewer for this comment. According to the suggestion of the reviewer, we have changed the term “collapse” to “quenching” in the revised manuscript (“In our case, the refractive index of Si will be modified significantly by injecting dense electron-hole plasma, which eventually leads to the quenching of the quasi-BIC (Supplementary Notes 6 and 7).”, see 2nd paragraph in page 7; “Since the injection of carriers can be completed in a short time (~100 fs), the quenching of the quasi-BIC following the thermalization of the injected carriers has no influence on the injection process.”, see 2nd paragraph in page 7). Similar descriptions have been used in the literature (see Refs. 44 and 45).

C10: What the Author can say about the polarization and directivity pattern of the generated light?

R10: We would like to thank the reviewer for this good question. In experiments, we examined the polarization of the luminescence emitted from Si cuboids by inserting a polarization analyzer in the signal collection channel. A typical example is shown in Fig. R2a. In Fig. R2b, we show the dependence of the luminescence intensity on the polarization angle observed at $\lambda = 600$ nm. It is found that the luminescence of the Si cuboid exhibits a linear polarization perpendicular to the length of the Si cuboid. We also simulated the three-dimensional (3D) radiation pattern of a Si cuboid ($l = 720$ nm, $w = 360$ nm) at $\lambda = 688$ nm, as shown in Fig. R2c. Also shown are the two-dimensional radiation patterns in the upper and lower backward focal planes. It is found that the radiation of the Si cuboid is governed by the radiations from ED and MQ, as shown in Fig. 2b. In the revised manuscript, we have added a brief discussion on this issue (“We examined the polarization of the luminescence emitted from Si cuboids by inserting a polarization analyzer in the collection channel. It was found that the luminescence of the Si cuboid exhibits a linear polarization perpendicular to the length of the Si cuboid. We also simulated the three-dimensional (3D) radiation pattern of a Si cuboid and found that the emission from the Si cuboid is governed by the radiations from ED and MQ (Supplementary Note 13).”, see 2nd paragraph in page 10). In addition, the detailed experimental and simulation results have been added in Supplementary Materials (see Supplementary Note 13).

Figure R2 (a) Luminescence spectra of a Si/SiO₂ cuboid ($l = 720$ nm, $w = 360$ nm) obtained by using a polarization analyzer with different polarization angles. (b) Dependence of the

luminescence intensity on the polarization angle observed at $\lambda = 600$ nm. (c) Three-dimensional radiation pattern simulated for a Si/SiO₂ cuboid ($l = 720$ nm, $l' = 530$ nm, $w = 360$ nm, $w' = 180$ nm, $h = 230$ nm, $h' = 180$ nm) at $\lambda = 688$ nm. Also shown are the two-dimensional radiation patterns in the upper and lower back focal planes.

Responses to the comments of the reviewer #2

(Manuscript ID: NCOMMS-21-32514-T)

We would like to thank the comments and suggestions of the reviewer which are definitely helpful for improving the quality of the manuscript. The responses (abbreviated as R) to these comments (abbreviated as C) and the changes made in the revised manuscript (marked in red color) are described in the following.

C1: Panmai and coauthors present a joint experimental and theoretical study of Si/SiO₂ cuboids, exploring the properties of a longitudinal quasi-BIC mode in these semiconductor cavities for the purpose of strongly enhancing the photoemission of Si. Given the sensitivity of the quasi-BIC mode to optical excitation, they focus on the response under nonlinear two-photon excitation achieved with ultrashort pulses. The experimental setup clearly shows a strong photoemission under high-power excitation targeting this mode, with properties supported by the theoretical data. The authors report a quantum efficiency above 13%. The manuscript and its complementing SI are very thorough in reporting the response of the system, characterizing its behavior through theoretical means, and discussing several divergences between some theoretical models and experiment. These results are, in my opinion, useful to continue the study and exploitation of Si systems in the development of future technology, and this manuscript should be of interests to many groups. Thus, conditional on the authors addressing some specific issues, I would recommend this manuscript for publication in Nature Communications.

R1: We appreciate the positive report of the reviewer.

C2: The quantum efficiency of the material is only compared with another system made by some of the same authors, using Si deposited over an Ag film and a thin SnO₂ spacer. Although this comparison is reasonable, specifically given that in both cases they used a similar method to estimate the quantum efficiency, and eminently useful, a broader overview of the literature would be expected to contextualize this value. This should also include systems with nominally larger quantum efficiencies, such as [DOI: 10.1103/PhysRevB.73.132302]. Then, it would be useful to actively discuss the important

features and differences between this work and others. As it stands, this work feels largely decontextualized.

R2: We would like to thank the reviewer for this comment. In indeed, the quantum efficiency is an important parameter characterizing the luminescence of a material and an accurate estimation of quantum efficiency is a key point. According to the suggestion of the reviewer, we have made a detailed survey for the quantum efficiencies of Si-based light sources, including porous Si and Si quantum dots (or nanocrystals) (see Supplementary Note 11). In general, the quantum efficiencies of Si nanocrystals are larger than those of porous Si. However, quantum efficiency values for Si nanocrystals scattered in a very wide range from ~10% to ~100%, depending strongly on the fabrication technique and the characterization method. In previously studies, the quantum efficiency was generally evaluated by seeking the ratio of the radiative recombination time to the nonradiative one. In these cases, the modification in the radiative recombination process induced by the Purcell effect or the local density of states need to be taken into account in order to obtain accurate values of quantum efficiency. In our case, we estimated the quantum efficiency of Si cuboids by measuring the numbers of the emitted and absorbed photons. The absorption of Si cuboids, which is dominated by nonlinear optical absorption at high pulse energies, offers us the opportunity to accurately extract the number of absorbed photons (see Supplementary Note 12). Thus, how to accurately estimate the number of emitted photons becomes a key point because only a fraction of emitted photons from a Si cuboid was detected. In this work, we simulated the collection efficiency of the emitted photons by considering both the directivity of the luminescence and the numerical aperture of the objective. An average collection efficiency of ~58 % was obtained by using this method, as shown in Fig. R1.

Fig. R1 Collection efficiencies (β) calculated for a Si/SiO₂ cuboid. ($l = 440$ nm, $l' = 340$ nm, $w = 260$ nm, $w' = 160$ nm, $h = 230$ nm, $h' = 180$ nm) at different emission wavelengths.

In the revised manuscript, we have added a detailed discussion on this important issue (“In general, the quantum efficiencies of Si nanocrystals are larger than those of porous Si. However, the quantum efficiency values for Si nanocrystals scattered in a very wide range from ~10% to ~100%, depending strongly on the fabrication technique and the characterization method (see Supplementary Note 11). In previously studies, the quantum efficiency was generally evaluated by seeking the ratio of the radiative recombination time to the nanoradiative one. In these cases, the modification in the radiative recombination process induced by the Purcell effect or the local density of states need to be taken into account in order to obtain accurate values of quantum efficiency. In our case, we estimated the quantum efficiency of Si cuboids by measuring the numbers of the emitted and absorbed photons. The absorption of Si cuboids, which is dominated by nonlinear optical absorption at high pulse energies, offers us the opportunity to accurately extract the number of absorbed photons (see Supplementary Note 12). Thus, how to accurately estimate the number of emitted photons becomes a key point because only a fraction of emitted photons from a Si cuboid was detected. In this work, we simulated the collection efficiency of the emitted photons by considering both the directivity of the luminescence and the numerical aperture of the objective. An average collection efficiency of ~58% was obtained by using this method (see

Supplementary Note 12). We also inspected the dependence of the excitation laser light reflected from the substrate (with the Si/SiO₂ cuboid) on the pulse energy (Fig. 4f). It is noticed that the optical absorption of the Si/SiO₂ cuboid, which is governed by linear absorption at low pulse energies, will become dominated by nonlinear absorption at high pulse energies. This unique feature offers us the opportunity to extract the number of absorbed photons from the deviation of the reflection intensity from the linear relationship observed at low pulse energies. The slope of this linear relationship can be calibrated by measuring the reflection intensities from the substrate only at different pulse energies. In this way, the quantum efficiency for the luminescence of the Si cuboid is found to be ~13% (Supplementary Note 12), which is further improved when comparing with the previous results for Si nanoparticles on an Ag film²²”, 2nd paragraph in page 9). In addition, a detailed survey on the quantum efficiencies of Si-based light sources, including the material types, preparation methods, quantum efficiencies, and measurement methods, has been added in Supplementary Materials (see Supplementary Note 11). Moreover, a detailed description on the estimation of the quantum efficiencies of Si/SiO₂ cuboids has been added in Supplementary Materials (see Supplementary Note 12).

C3: The method for quantifying the quantum efficiency depends strongly on the linear fit shown in Fig. 5f and S13, so the reported quantum efficiency is sensitive to the specific details and assumptions behind this fit, and they should be discussed in more detail. For instance, in selecting the initial linear trend, what constitute “low pulse energies”? Theoretically, what is the expected behavior of the reflection with respect to pulse energy in the presence of the cuboids? Would it be possible, and useful, to obtain such data without the cuboids and use it as the baseline, instead of the linear fit?

R3: We would like to thank the reviewer for this good question. Indeed, it is very important to provide an estimation of the quantum efficiency as accurate as possible for the luminescence of a Si cuboid. In practice, it remains a big challenge to evaluate the quantum efficiency for the luminescence of a single nanoparticle from the experimental point of view. Basically, the quantum efficiency for the luminescence of a nanoparticle is defined as the ratio of the number of emitted photons to that of absorbed photons. While the former can be extracted by

analyzing the luminescence spectrum, the latter is generally difficult to estimate. Fortunately, the Si cuboids investigated in this work are excited by femtosecond laser pulses and the nonlinear optical absorption is dominated at high pulse energies, offering us the opportunity to estimate the number of absorbed photons by exploiting the nonlinear dependence of the reflected laser light on the pulse energy, i.e., the deviation of the intensity of the reflected laser light at high pulse energies from the linear relationship observed at low pulse energies. The principle of this method as well as its practical application in estimating the quantum efficiency of Si nanoparticles has been described in detail in previous studies (see Supplementary Note 12 of Ref. 24).

As indicated by the reviewer, the accuracy of this method relies on the determination of the linear relationship between the reflected laser light intensity and the pulse energy. It should be emphasized that we measured the reflection signal of a laser light, which is much stronger than any luminescence, by using a highly sensitive EMCCD. A band-stop filter with a large attenuation coefficient ($\sim 10^{-6}$) must be used to attenuate the reflected light in order to avoid the damage of the EMCCD. Therefore, the dependence of the reflected laser light intensity on the excitation pulse energy could be accurately determined even at low pulse energies. In Fig. R2a, we present the dependence of the reflected laser light intensity on the pulse energy measured in the cases with and without a Si cuboid ($l = 440$ nm, $w = 260$ nm). In both cases, a linear relationship is observed. It is noticed that the slopes of the two linear relationships are almost the same and the reflectivity is slightly larger in the presence of the Si cuboid. The linear relationship remains even at high pulse energies in the absence of the Si cuboid. Therefore, the slope in the presence of the Si cuboid obtained at low pulse energies can be calibrated by using the value obtained in the absence of the Si cuboid. We also performed numerical simulations for the reflection spectra from a sapphire substrate with and without a Si/SiO₂ cuboid ($l = 540$ nm, $l' = 350$ nm, $w = 340$ nm, $w' = 150$ nm, $h = 230$ nm, $h' = 180$ nm), as shown in Fig. R2b. It is remarkable that the reflectivity at the femtosecond laser light (720 nm) is higher in the presence of the Si/SiO₂ cuboid.

Fig. R2 (a) Dependence of the intensity of the reflected laser light on the pulse energy measured for a sapphire substrate without and with a Si/ SiO₂ cuboid ($l = 440$ nm, $l' = 340$ nm, $w = 260$ nm, $w' = 160$ nm, $h = 230$ nm, $h' = 180$ nm). (b) Reflection spectra simulated for a sapphire substrate ($d = 50000$ nm) without and with a Si/ SiO₂ cuboid ($l = 440$ nm, $l' = 340$ nm, $w = 260$ nm, $w' = 160$ nm, $h = 230$ nm, $h' = 180$ nm).

In the revised manuscript, we have added a brief discussion on this issue in the main text (“We also inspected the dependence of the excitation laser light reflected from the substrate (with the Si/SiO₂ cuboid) on the pulse energy (Fig. 4f). It is noticed that the optical absorption of the Si/SiO₂ cuboid, which is governed by linear absorption at low pulse energies, will become dominated by nonlinear absorption at high pulse energies. This unique feature offers us the opportunity to extract the number of absorbed photons from the deviation of the reflection intensity from the linear relationship observed at low pulse energies. The slope of this linear relationship can be calibrated by measuring the reflection intensities from the substrate only at different pulse energies. In this way, the quantum efficiency for the luminescence of the Si cuboid is found to be ~13% (Supplementary Note 12), which is further improved when comparing with the previous results for Si nanoparticles on an Ag film²².”, see 2nd paragraph in page 9) and provided a detailed description for the estimation of the quantum efficiencies of Si/SiO₂ cuboids in Supplementary Materials (see Supplementary Note 12).

C4: Overall, I think that the methodology is not reported in sufficient detail. This is so for the experiment, but most especially for the theory.

R4: According to the suggestion, we have added a schematic to interpret in detail the physical mechanism for the generation of highly efficient white light from Si cuboids by exploiting the enhanced nonlinear optical absorption mediated by the quasi-BICs supported in Si cuboids (**“Physical mechanism for realizing efficient nonlinear optical emission from Si/SiO₂ cuboid**

In Fig. 1d, we illustrate the physical mechanism for realizing efficient nonlinear optical emission from Si/SiO₂ cuboids by exploiting the significantly enhanced nonlinear optical absorption achieved at the quasi-BICs supported by Si/SiO₂ cuboids. Basically, high-density electron-hole pairs can be generated in a Si/SiO₂ cuboid via two- or three-photon-induced absorption (2PA or 3PA) upon the excitation of the Si/SiO₂ cuboid by using femtosecond laser pulses. Since the 2PA or 3PA is proportional to the fourth or sixth power of the electric field inside the Si cuboid (i.e., $|E|^4$ or $|E|^6$), a significantly enhanced 2PA or 3PA is expected at the quasi-BIC where the maximum electric field (or quality factor) is achieved. In general, the hot electrons generated in the conduction band of Si will relax rapidly (less than 1.0 ps) from the Γ point to the Δ point via the emission of phonons and then recombine radiatively with the holes in the valence band with the help of phonons. However, such relaxation and recombination processes will be greatly alleviated in the high-density case by the Auger process, which is proportional to the cubic of the carrier density. As a result, the electrons remain “hot” at the high-energy states around the Γ point, which increases dramatically the relaxation time by two orders of magnitude (~50 ps) or the possibility for the vertical transition to the valence band. The enhanced electric field achieved at the Mie resonances of the Si/SiO₂ cuboid, such as electric and magnetic dipoles and quadrupoles (ED, MD, EQ, and MQ etc.), will accelerate the radiative recombination of hot electrons via the Purcell effect (~500 ps). Moreover, the radiative recombination time becomes inversely proportional to the carrier density at high-density case, leading to highly efficient nonlinear optical emission from the Si/SiO₂ cuboid. On the other hand, the Si/SiO₂ cuboid could be heated to a high temperature due to the thermalization of hot electrons. A sufficiently high temperature (e.g., ~1500 K) will trigger the intrinsic excitation of carriers in Si, which supplies a huge number

of electrons from the valence band to the Δ point of the conduction band²². In this case, a significantly enhanced nonlinear optical emission is expected, which is manifested in the burst of hot electron luminescence. It should be emphasized that the key point of this scenario is the generation of high-density carriers, which is greatly enhanced by exploiting the quasi-BIC supported by the Si/SiO₂ cuboid, as demonstrated in this work.”, see Fig. 1d and the description in the main text, 2nd paragraph in page 5). In addition, we have provided more information on the experimental aspects, especially a detailed description for the estimation of the quantum efficiencies of Si/SiO₂ cuboids (see Supplementary Note 12).

Fig. 1 | Structure and morphology of Si/SiO₂ cuboids supporting quasi-BICs and the physical mechanism for efficient white light emission. **a**, Schematic showing a Si cuboid supported by a sapphire (Al₂O₃) substrate. Also shown is the magnetic field distribution at the quasi-BIC. **b**, Schematic showing the detailed structure of a Si/SiO₂ cuboid supported by a sapphire substrate, which is defined by geometrical parameters of l , w , h , l' , w' , h' , and t . **c**, SEM image of a regular array of Si/SiO₂ cuboids. **d**, Schematic showing the physical mechanism for efficient white light emission from Si/SiO₂ cuboids by exploiting the enhanced nonlinear optical absorption at quasi-BICs.

C5: I believe that the similarity between Fig. 3b,c (theory) and Fig. 3d,e (experiment) is significantly overstated. The connection should be further discussed.

R5: We would like to thank the reviewer for carefully reading the manuscript. In Fig. 3b,c, we present the scattering and TPA spectra calculated for two Si cuboids with different lengths ($l = 740$ nm and $l = 780$ nm). The scattering and TPA spectra measured for two Si/SiO₂ cuboids with similar structure parameters are provided in Fig. 3d,e. The discrepancies between the simulation results and the experimental observations are observed mainly in the scattering spectra and they are caused by two reasons. One is the influence of the SiO₂ layers on the surfaces of Si cuboids and the other is the lower quantum efficiency of the detector at long wavelengths (> 850 nm). In the revised manuscript, we have provided simulation and measurement results for Si/SiO₂ cuboids (i.e., the SiO₂ layer on the surfaces of Si cuboids are taken into account) which exhibit smaller discrepancies (see Fig. 3 in the revised manuscript). In addition, we have added a brief discussion on this issue (“The discrepancies between the simulation results and the experimental observations are observed mainly in the scattering spectra and they are caused by two reasons. One is the influence of the SiO₂ layers on the surfaces of Si cuboids and the other is the lower quantum efficiency of the detector at long wavelengths (>850 nm).”, see 1st paragraph in page 8).

C6: It could be informative to expand on the qualitative differences between theory and experiment in Fig. 2. Although the anisotropy discussed in the SI does explain the shift in the q-BIC, the experimental scattering presents some overall different features to the theory. Would it be useful to recreate Fig. 2b with the anisotropic Si/SiO₂ cuboid?

R6: We would like to thank the reviewer for this good question. According to the suggestion of the reviewer, we have used the asymmetric oxidation model described in Supplementary Materials (see Supplementary Note 4) to simulate the scattering spectrum shown in Fig. 2b. By appropriately choosing the thicknesses of the SiO₂ layers in different directions (x, y, and z directions), we were able to obtain a scattering spectrum which agrees well with the experimental observation. The parameters used in the numerical simulation are provided in Fig. 2b. In the revised manuscript, we have provided new simulation results for a Si/SiO₂ cuboid in Fig. 2b and modified the description for Fig. 2b accordingly (“By considering

anisotropic oxidation of Si cuboids during the fabrication process, we can obtain a scattering spectrum in which the quasi-BIC agrees well with that observed in the experiment (Fig. 2b).”

see 2nd paragraph in page 6).

Responses to the comments of the reviewer #3

(Manuscript ID: NCOMMS-21-32514-T)

We would like to thank the comments and suggestions of the reviewer which are definitely helpful for improving the quality of the manuscript. The responses (abbreviated as R) to these comments (abbreviated as C) and the changes made in the revised manuscript (marked in red color) are described in the following.

C1: In this work, the authors describe Si cuboids as efficient emitters of white light upon two-photon excitation of a hot electron plasma. They show that cuboids with specific dimensions can sustain bound states in the continuum upon pulsed femtosecond optical pumping due to good overlap of the pump pulse with the optical resonances that are sustained in the cuboid. By matching the cuboid size to a resonance that is slightly larger in frequency than half on the Si band gap, a strong hot electron plasma is achieved that leads to white light emission. This excitation is more efficient with TE than TM modes. The overlap of optical modes in the cuboid with the TPA and the line shapes of the scattering spectra are discussed in great detail and the reasoning is convincing. Overall, the work is very interesting and timely.

R1: We appreciate the positive report of the reviewer.

C2: However, for publication in Nature Comm, I would request a deeper discussion of the process underlying the light emission. What are the involved timescales here? How can the profile of the emission spectrum be understood? I suggest to include a scheme that illustrates the excitation and creation of the plasma and the channels for relaxation (bright and dark). Can an intuitive picture for the emission process be provided? A figure giving a schematic description would be valuable.

R2: We would like to thank the reviewer for this valuable suggestion. According to the suggestion of the reviewer, we have provided an intuitive picture to illustrate the excitation and emission processes involved in the white light emission of Si cuboids (see Fig. 1d in the main text). In addition, we have provided the timescales for the relaxation times at low- and

high-density cases and the radiative recombination times at high-density cases. (“**Physical mechanism for realizing efficient nonlinear optical emission from Si/SiO₂ cuboid**”

In Fig. 1d, we illustrate the physical mechanism for realizing efficient nonlinear optical emission from Si/SiO₂ cuboids by exploiting the significantly enhanced nonlinear optical absorption achieved at the quasi-BICs supported by Si/SiO₂ cuboids. Basically, high-density electron-hole pairs can be generated in a Si/SiO₂ cuboid via two- or three-photon-induced absorption (2PA or 3PA) upon the excitation of the Si/SiO₂ cuboid by using femtosecond laser pulses. Since the 2PA or 3PA is proportional to the fourth or sixth power of the electric field inside the Si cuboid (i.e., $|E|^4$ or $|E|^6$), a significantly enhanced 2PA or 3PA is expected at the quasi-BIC where the maximum electric field (or quality factor) is achieved. In general, the hot electrons generated in the conduction band of Si will relax rapidly (less than 1.0 ps) from the Γ point to the Δ point via the emission of phonons and then recombine radiatively with the holes in the valence band with the help of phonons. However, such relaxation and recombination processes will be greatly alleviated in the high-density case by the Auger process, which is proportional to the cubic of the carrier density. As a result, the electrons remain “hot” at the high-energy states around the Γ point, which increases dramatically the relaxation time by two orders of magnitude (~50 ps) or the possibility for the vertical transition to the valence band. The enhanced electric field achieved at the Mie resonances of the Si/SiO₂ cuboid, such as electric and magnetic dipoles and quadrupoles (ED, MD, EQ, and MQ etc.), will accelerate the radiative recombination of hot electrons via the Purcell effect (~500 ps). Moreover, the radiative recombination time becomes inversely proportional to the carrier density at high-density case, leading to highly efficient nonlinear optical emission from the Si/SiO₂ cuboid. On the other hand, the Si/SiO₂ cuboid could be heated to a high temperature due to the thermalization of hot electrons. A sufficiently high temperature (e.g., ~1500 K) will trigger the intrinsic excitation of carriers in Si, which supplies a huge number of electrons from the valence band to the Δ point of the conduction band²². In this case, a significantly enhanced nonlinear optical emission is expected, which is manifested in the burst of hot electron luminescence. It should be emphasized that the key point of this scenario is the generation of high-density carriers, which is greatly enhanced by exploiting the

quasi-BIC supported by the Si/SiO₂ cuboid, as demonstrated in this work.”, see Fig. 1d and the description in the main text).”, see 2nd paragraph in page 5).

Fig. 1 | Structure and morphology of Si/SiO₂ cuboids supporting quasi-BICs and the physical mechanism for efficient white light emission. **a**, Schematic showing a Si cuboid supported by a sapphire (Al₂O₃) substrate. Also shown is the magnetic field distribution at the quasi-BIC. **b**, Schematic showing the detailed structure of a Si/SiO₂ cuboid supported by a sapphire substrate, which is defined by geometrical parameters of l , w , h , l' , w' , h' , and t . **c**, SEM image of a regular array of Si/SiO₂ cuboids. **d**, Schematic showing the physical mechanism for efficient white light emission from Si/SiO₂ cuboids by exploiting the enhanced nonlinear optical absorption at quasi-BICs.

C3: Does the emission stem from transitions in between electronic levels of the Si, or does it come from electrical discharge (maybe generated by displacement charges)?

R4: We would like to thank the reviewer for this good question. Actually, we have sufficient experimental results to verify that the white light emission from Si cuboids originates from the transitions between the electronic levels of Si, rather than electric discharge. In our

previous studies (Refs. 18 and 22), we have presented a detailed explanation for the physical mechanism responsible for the white light emission from Si nanoparticles. In experiments, we observed enhanced luminescence at the MQ/EQ resonances of a Si nanoparticle when the MD resonance is resonantly excited. Similarly, the luminescence enhancement at the MD resonance was also revealed when the MQ/EQ resonance of the Si nanoparticle was resonantly excited (Ref. 20). In addition, the dependence of the luminescence intensity on the excitation irradiance also exhibited clearly two- and three-photon-induced absorption which gives a slope of 2 and 3 (see Supplementary Note 4 in Ref. 18 and Supplementary Note 15 in Ref. 22, we reproduced the result in Fig. R1 for the convenience of the reviewer).

Fig. 3 Emission enhancement and luminescence lifetime. **a** Nonlinear response spectra measured at different excitation pulse energies for a silicon nanosphere (NS) with $d \sim 210$ nm. The spectrum of I calculated for the silicon NS is also presented for comparison. The inset shows the dependence of the up-converted luminescence on the excitation pulse energy plotted in a double-logarithmic coordinate. **b** Decay of the up-converted luminescence measured for a silicon NS with $d \sim 190$ nm after the excitation of the femtosecond laser pulses. The luminescence lifetime is derived to be ~ 52 ps based on a reconvolution fitting analysis. Here, IRF represents instrument response function and χ^2 is a parameter characterizing the fitting quality. **c** The corresponding residuals for the fitting

Enhanced luminescence observed at the MQ/EQ resonances. [Ref. 18: *Nat. Commun.* **9**, 2964 (2018)]

FIG. 3. The MD resonance shift revealed in the MD-enhanced hot luminescence. (a) Hot-electron luminescence spectra measured for a Si nanosphere excited at different wavelengths of 540, 570, and 600 nm. The scattering spectrum measured for the Si nanosphere is also provided for reference. (b) The dependence of the PL intensity and MD resonance on the excitation wavelength.

Enhanced luminescence observed at the MD resonance. [Ref. 20: *Phys. Rev. Applied* **13**, 014003 (2020)]

Supplementary Figure 16. Excitation power density dependent luminescence spectra and wavelength dependent extracted slopes measured for three silicon nanospheres. They were resonantly excited at their magnetic dipole resonances located at (a) 775 nm, (c) 750 nm, and (e) 730 nm. The corresponding dependences of the extracted slope on the wavelength of the emitted photon are shown in (b), (d), and (f), respectively.

Two- and three-photon-induced absorption of single Si nanoparticles. [Ref. 18: *Nat. Commun.* **9, 2964 (2018)]**

Figure S17 (a) Luminescence spectra measured for a Si nanoparticle excited at different powers. (b) Slopes extracted from the dependence of the luminescence intensity on the laser power plotted in a logarithmic coordinate at different wavelengths.

Two- and three-photon-induced absorption of single Si nanoparticles on an Ag film. [Ref. 22: *Nano Lett.* **21**, 2397–2405 (2021)]

Fig. R1 Enhanced luminescence observed at the Mie resonances (EQ/MQ, MD) of a Si nanoparticle and the dependence of the luminescence intensity on the excitation irradiance (reproduced from Refs. 18, 20, and 22).

The experimental observations that undoubtedly exclude electrical discharge are described in detail in the following (see R5, response to C5). To address this comment, we have added a brief discussion on this issue in the revised manuscript (“It should be emphasized that the white light emission from Si nanoparticles, including Si cuboids studied in this work, originates from the interband transition of hot electrons, rather than other physical origins such as electrical discharge. Previously, the enhanced hot electron luminescence from a Si nanoparticle was observed at the MQ/EQ resonances or MD resonance of the Si nanoparticle^{18,20}. In addition, the dependence of the luminescence intensity on the excitation irradiance exhibits a slope in between 2.0 and 3.0, varying the 2PA/3PA process involved in the luminescence¹⁸. Moreover, the scattering spectra of Si/SiO₂ cuboids remain unchanged before and after the luminescence burst, implying no change in the crystalline structure (Supplementary Note 10). Finally, the hot electron luminescence from Si cuboids was not

observed when the high-Q quasi-BICs were resonantly excited by using picosecond laser pulses²⁹. All these experimental observations indicate undoubtedly that the luminescence from Si cuboids belongs to nonlinear optical emission originating from the interband transition of hot electrons.”, see 2nd paragraph in page 8).

C4: Here, the spectral profile at or just below the threshold (like Fig 4e) would be interesting to see.

R4: According to the suggestion of the reviewer, we have examined the luminescence spectra of Si/SiO₂ cuboids below and above the threshold. A typical example is shown in Fig. R2. It was found that several peaks corresponding to the enhanced luminescence at the Mie resonances of the Si cuboid (e.g., ED/EQ/MQ) were observed for pulse energies just below (0.82 pJ) and above the threshold (0.90 pJ). In the revised manuscript, we have modified the description of Fig. 4e (“The luminescence spectra below and above the threshold appear as broadband emissions with enhancements observed at the Mie resonances of the Si cuboid (Supplementary Note 9.2).”, see 2nd paragraph in page 8). The experimental results have been added in Supplementary Materials (see Supplementary Note 9.2).

Fig. R2 Luminescence spectra measured for a Si/SiO₂ cuboid with $l = 440$ nm and $w = 260$ nm before (a) and after (b) the threshold pulse energy which is 0.90 pJ.

C5: Could it be that optically induced displacement charges are involved in the white emission? In the sense that a strong field is build up between the two ends of the cuboid that

leads to electrical discharge generating the white light?

R5: We would like to thank the reviewer for this good question. Actually, it is well known that glow-discharge occurs when a high voltage is applied on the two electrodes of a chamber containing a low-pressure gas, generating white light. Similarly, white light emission is also observed when the voltage applied on a thin polymer exceeds a threshold. In these cases, the physical mechanism for the generation of white light is ascribed to electrical discharge, in which the ionization of atoms or molecules occurs. It should be emphasized that a static voltage with sufficient high value is applied on gas or polymer in order to induce the ionization of atoms or molecules that generates white light. For oscillating electric field at optical frequencies, it is known that terahertz waves can be generated by ionization of air induced by high-power femtosecond laser pulses. In addition, supercontinuum with broadband can be generated by irradiating water with high-power femtosecond laser pulses. However, the underlying physical mechanism in these cases is nonlinear optics rather than electrical discharge.

In our case, we excited Si cuboids by using femtosecond laser pulses with low powers. If the electrical discharge occurs, the crystalline structure of Si will be damaged at least to some extent. However, we observed the stable white light emission and repeated luminescence burst from Si cuboids. More importantly, we examined the scattering spectra of Si cuboids before and after the luminescence burst and found no difference between them, as shown in Fig. R3. It indicates clearly that there is no change in the crystalline structure (or refractive index) of Si, which is not expected if electrical discharge occurs. Previously, we also excited Si cuboids by using picosecond laser pulses at $\sim 1.55 \mu\text{m}$ (see Ref. 29). In this case, the oscillating electric field applied on Si cuboids is similar or even larger than that achieved by using femtosecond laser pulses at $\sim 800 \text{ nm}$. However, we did not observe white light emission from Si cuboids. Instead, we observed efficient second and third harmonic generation from Si cuboids. This behavior indicates that electrical discharge induced by the oscillating electric field is not the physical origin for the white light emission of Si cuboids.

In the revised manuscript, we have added a brief discussion on this issue (“**It should be emphasized that the white light emission from Si nanoparticles, including Si cuboids studied in this work, originates from the interband transition of hot electrons, rather than other**

physical origins such as electrical discharge. Previously, the enhanced hot electron luminescence from a Si nanoparticle was observed at the MQ/EQ resonances or MD resonance of the Si nanoparticle^{18,20}. In addition, the dependence of the luminescence intensity on the excitation irradiance exhibits a slope in between 2.0 and 3.0, varying the 2PA/3PA process involved in the luminescence¹⁸. Moreover, the scattering spectra of Si/SiO₂ cuboids remain unchanged before and after the luminescence burst, implying no change in the crystalline structure (Supplementary Note 10). Finally, the hot electron luminescence from Si cuboids was not observed when the high-Q quasi-BICs were resonantly excited by using picosecond laser pulses²⁹. All these experimental observations indicate undoubtedly that the luminescence from Si cuboids belongs to nonlinear optical emission originating from the interband transition of hot electrons.”, see 2nd paragraph in page 8). In addition, the scattering spectra measured for a Si/SiO₂ cuboid before and after the luminescence burst have been added in Supplementary Materials (see Supplementary Note 10).

Fig. R3 Scattering spectra measured for a Si/SiO₂ cuboid with $l = 600$ nm and $w = 280$ nm before and after the luminescence burst.

C6: Time resolved optical spectroscopy would be very interesting, both transient absorption and the PL decay traces to get deeper insight into the emission process.

R6: We agree with the reviewer that deeper insight into the emission can be obtained by using time-solved optical spectroscopy, such as transient absorption spectra and PL decay traces suggested by the reviewer. Basically, the transient absorption spectra for Si cuboids can be achieved by using the so-called pump-probe technique. In this case, the probe pulse should be a supercontinuum with a broadband covering the visible to near infrared spectral range, which is generated by femtosecond laser pulses. Since the optical characterizations of Si cuboids are performed under a microscope, such a pump-probe measurement remains a big challenge at present although we are trying to build such an optical system.

To address this comment, we have calculated the absorption spectra for a Si cuboid ($l' = 560$ nm, $w' = 180$ nm, and $h' = 180$ nm) at different injected carrier densities, as shown in Fig. R4. The refractive indices at different carrier densities were derived by using the method described in Ref. 20. In each case, one can identify four absorption peaks in the absorption spectrum. With increasing carrier intensity, a blueshift is observed for all absorption peaks. For the absorption peak corresponding to the quasi-BIC ($\lambda \sim 760$ nm), the absorption increases initially with increasing carrier density and reaches a maximum rapidly at 2.0×10^{20} cm⁻³. Then, it begins to decrease with increasing carrier density. The reduction of the absorption is caused by the quenching of the quasi-BIC at high doping levels (see Refs. 43 and 44). In contrast, a rapid increase in absorption is observed for the other three peaks and the saturation of absorption is observed at high doping levels.

In experiments, we have performed luminescence decay measurements for Si cuboids. A typical example obtained for a Si/SiO₂ cuboid ($l = 600$ nm, and $w = 340$ nm) is shown in Fig. R5. The luminescence lifetimes extracted by fitting the luminescence decays are found to be ~ 110 ps and ~ 49 ps before and after the luminescence burst.

In the revised manuscript, we have added a brief discussion on this issue (“Basically, the transient absorption spectra for Si/SiO₂ cuboids can be achieved by using the so-called pump-probe technique. In this case, a supercontinuum with a broadband covering the visible to near infrared spectral range, which is generated by femtosecond laser pulses, is necessary. Since the optical characterizations of Si cuboids are performed under a microscope, such a

pump-probe measurement remains a big challenge at present. As an alternative, we calculated the absorption spectra of a Si cuboid at different injected carrier densities. The quenching of the quasi-BIC was observed at high carrier densities (see Supplementary Note 6). We also measured the luminescence lifetimes of Si/SiO₂ cuboids. It was found that the luminescence lifetime, which is ~111 ps below the threshold, is reduced to be ~49 ps after the luminescence burst (see Supplementary Note 14).”, see 3th paragraph in page 10). In addition, the simulation results for transient absorption spectra and the experimental results for the luminescence lifetimes have been added in Supplementary Materials (see Supplementary Notes 6 and 14).

Fig. R4 Evolution of the absorption spectrum with increasing carrier density calculated for a Si cuboid with $l' = 560$ nm, $w' = 180$ nm, and $h' = 180$ nm.

Fig. R5 Luminescence decays measured for a Si/SiO₂ cuboid ($l = 600$ nm, $w = 340$ nm) at $\lambda = 720$ nm before (a) and after (b) the luminescence burst. In each case, the luminescence lifetime is extracted by fitting the luminescence decay with an exponential function.

C7: I have difficulties to follow the transition from the Fano line shape to the Lorentzian on in Figure 4c. Can the authors show more line plots?

R7: Here, we guess that the reviewer talked about the evolution of the scattering spectra shown in Fig. 2c or Fig. 3b,c (not Fig. 4c). Basically, the Fano lineshape observed in the scattering spectrum of a Si cuboid originates from the interference of two optical modes and it can be characterized by using an asymmetry parameter q . By varying the geometrical parameter of the Si cuboid (such as its length), the phase difference between the two optical modes is changed, leading to the variation of the Fano lineshape and the value of q . A symmetric Lorentz lineshape is obtained when $q \rightarrow \infty$, corresponding to the quasi-BIC. A transition of the q value from negative to positive occurs at the BIC point, which is manifested in the change of the Fano lineshape. According to the suggestion of the reviewer, we have provided a typical example showing the evolution of the scattering spectrum calculated for Si/SiO₂ cuboids with different lengths, as shown in Fig. R6. Based on the fitting of the scattering spectrum, a transition of the lineshape from Fano to Lorentzian is clearly identified. Similar behavior was also described in detail in Ref. 41. In the revised manuscript, we have provided scattering spectra measured for Si/SiO₂ cuboids with different structure parameters. Although the measured scattering spectra exhibit some discrepancies as compared with the simulated ones, we can still observe the evolution of the Fano lineshape and the appearance of the Lorentz lineshape (see Supplementary Note 5.2).

Fig. R6 Scattering spectra calculated for Si/SiO₂ with different structure parameters.

C8: The threshold value for the pulse energy on top of page 9 does not match the threshold in Figure 4e

R8: We would like to thank the reviewer for carefully reading the manuscript. We are sorry for the mismatch between the description in the main text and that shown in Fig. 4e because Si cuboids with different sizes have different thresholds. We have corrected this error in the revised manuscript (“The Si/SiO₂ cuboid ($l = 440$ nm and $w = 260$ nm) located at the center of the image was excited by using femtosecond laser pulses with a pulse energy of $E = 0.82$ pJ, emitting hot electron luminescence (Fig. 4c). In this case, the Si/SiO₂ cuboid appeared as a bright spot in the image. Surprisingly, we observed the burst of luminescence when the excitation pulse energy was raised to $E_{th} = 0.90$ pJ.”, see 2nd paragraph in page 8).

C9: On page 3 in the introduction the authors relate increased Auger processes to increase in quantum efficiency. That is very unusual, as far as I know Auger processes are detrimental for radiative recombination. Can the authors elaborate on this?

R9: We would like to thank the reviewer for this good question. As discussed in R2, the Auger process plays a crucial role in increasing the relaxation time of hot electrons (or keeping the electrons “hot” round the Γ point, increasing the possibility for the direct bandgap transition from the Γ point to the valence. As a result, the quantum efficiency, which is defined as $\eta = 1/(1+\tau_r/\tau_{nr})$. This strategy of utilizing the Auger process to enhance the quantum efficiencies of semiconductors with indirect bandgaps has been described in our previous studies (see Refs. 18 and 22). As schematically shown in Fig. 1d, the electrons in the valence band of Si are lifted to the conduction band (Γ point) via highly-efficient two- and three-photon-induced absorption (2PA and 3PA) mediated by the quasi-BIC, generating high-density electron-hole plasma in the Si cuboid. Hot electrons in the conduction band will relax from the Γ point to the Δ point via the emission of phonons. However, the radiative transition of electrons from the Δ point to the valence band is hindered by the Auger process, which lifts the electrons at the Δ point to the high-energy states around the Γ point. Therefore, the Auger process is detrimental to the radiative recombination of carriers from the Δ point to the valence band, which is indirect bandgap transition. It is remarkable, however, that

relaxation time of hot electrons from the Γ point to the Δ point is dramatically increased by the Auger process, which become significant at high carrier densities (proportional to the cubic of the carrier density). Consequently, the direct bandgap transition of hot electrons from the Γ point to the valence band, which may facilitated by the Mie resonances of the Si cuboid (e.g., MD/ED and MQ/EQ), is greatly enhanced. This strategy has been previously exploited to light up Si nanoparticles (see Refs. 18 and 22). In the revised manuscript, we have added a section to interpret the physical mechanism for the efficient nonlinear optical emission from Si cuboids mediated by quasi-BICs (see the details in R2).

C10: English language needs improvements

R10: We have tried our best to improve the English writing with the help of native speakers.

C11: Overall, manuscript needs a careful correction for typos and inconsistencies, for example the caption of Fig. 2 has 3 typos, and a confusion on the value of w . Also in Figure S11 the l and w labels are not clear.

R11: We would like to than the reviewer for carefully reading the manuscript. We are very sorry for the mistakes and typos appearing in the manuscript. In the revised manuscript, we have corrected these mistakes and typos.

C12: In summary, I can recommend publication after these points are addressed.

R12: We have responded point-by-point the comments of the reviewer and revised the manuscript based on the comments and suggestions of the reviewer. We appreciate again the effort made by the reviewer to improve the quality of the manuscript and hope that the manuscript after revision will fulfill the criterion for publication in Nature Communications.

REVIEWER COMMENTS

Reviewer #1 (Remarks to the Author):

I would like to thank the Authors for addressing all the questions and comments. The manuscript has been substantially improved and I'm pleased to recommend it for publication.

Reviewer #2 (Remarks to the Author):

I would like to thank the authors for their careful responses to my recommendations. I think that their edits have improved the clarity of the discussion, and the manuscript is ready for publication.

Just a couple of minor final comments:

1) When I mentioned that the theoretical methodology was not reported in sufficient detail, I was not referring to the theory not being explained clearly (although the new Fig. 1d is useful nonetheless), but the fact that the simulation methodology was not reported. As far as I have seen, the manuscript simply mentions using COMSOL and Lumerical, without specifying why using both, what systems were created with them, and in which configurations. Ideally, there should be enough information so that other groups could also replicate the simulations.

2) I am confused by the definition of external quantum efficiency noted in the SI: "the number of emitting photons to that of detected photons". This strikes me as non-standard, and perhaps I am misunderstanding it. What do you mean by "detected photons"? I was expecting to see the EQE characterized as the ratio of emitted photons to photons impinging on the device.

Additionally, it would be worth revisiting the literature cited in the table, for accuracy. For instance, browsing a random paper from the table, Ref. 7, it seems to me that what they report are internal quantum efficiency, not external.

Reviewer #3 (Remarks to the Author):

In this very thorough revision the authors have ratified my comments, and I can recommend publication.

Responses to the comments of the reviewer #2

(Manuscript ID: NCOMMS-21-32514A)

We would like to thank the effort made by the reviewer for improving the quality of the manuscript. The responses (abbreviated as R) to the comments/suggestions (abbreviated as C) of the reviewer and the changes we made in the revised manuscript (marked in red color) are described in the following.

C1: I would like to thank the authors for their careful responses to my recommendations. I think that their edits have improved the clarity of the discussion, and the manuscript is ready for publication.

R1: We would like to thank the reviewer for the positive report and recommendation.

C2: When I mentioned that the theoretical methodology was not reported in sufficient detail, I was not referring to the theory not being explained clearly (although the new Fig. 1d is useful nonetheless), but the fact that the simulation methodology was not reported. As far as I have seen, the manuscript simply mentions using COMSOL and Lumerical, without specifying why using both, what systems were created with them, and in which configurations. Ideally, there should be enough information so that other groups could also replicate the simulations.

R2: We are sorry for misunderstanding the meaning of the reviewer for methodology. As indicated by the reviewer, we did not provide a detailed description for the numerical simulations, which can be used for the readers to replicate the simulation results. In this work, we used two methods (finite element method and FDTD method) to perform numerical simulations. For the finite element method, the Maxwell equations are solved in frequency domain. In comparison, the Maxwell equations are solved in time domain when the FDTD method is employed. Both of them can be used to calculate the scattering spectra of Si/SiO₂ cuboids and the electric/magnetic field distributions in them. However, we can easily calculate the integration of the electric field over a Si/SiO₂ cuboid when the finite element method is used, which characterizes the nonlinear optical absorption of the Si/SiO₂ cuboid. On the other hand, we can readily extract the quality factor of the Si/SiO₂ cuboid by monitoring the decay of the electric field in it when the FDTD method is employed. Therefore

we used both of them in the numerical simulations and very good agreements between the simulation results were obtained when these two methods were used to calculate the scattering spectra and field distributions. According to the suggestion of the reviewer, we have added a detailed description for the numerical simulations in the revised manuscript, including the advantages of the two methods, the configurations of the simulated objects, the refractive indices of the materials and surrounding medium, the volume of the simulation region, the shape and size of the mesh, and the boundary condition (“The scattering spectra of Si and Si/SiO₂ cuboids and the corresponding electric and magnetic field distributions were calculated based on the finite element method (Multiphysics, COMSOL) and the finite-difference time-domain method (FDTD solution). Although the Maxwell equations were solved in frequency and time domains respectively, very good agreements were found between the simulation results obtained by using these two methods. By using the finite element method, we could easily derive the integration of the electric field over a Si/SiO₂ cuboid (e.g., $\int |E(\lambda)|^4 dV/V$), which characterizes the nonlinear optical absorption of the Si/SiO₂ cuboid. On the other hand, the decay of electric field inside a Si/SiO₂ cuboid, which gives the Q factor of the Si/SiO₂ cuboid, could be readily obtained by using the FDTD method.

In the numerical simulations, the height of Si cuboids was fixed to be $h = 180$ nm while the length and width of Si cuboids were varied in order to find out the quasi-BIC modes suitable for the excitation of Si/SiO₂ cuboids. This height corresponds to the thickness of the Si layer in the SOS wafer used for the fabrication of Si cuboids after the thermal oxidation (see more details in Supplementary Note 1). The refractive index of Si was taken from Aspnes⁵⁶ while those of SiO₂ and Al₂O₃ were chosen to be 1.45 and 1.70. In each case, the Si/SiO₂ cuboid located on an Al₂O₃ substrate was placed at the center of the simulation region, which was enclosed by a perfectly matched layer that absorbs completely the outgoing light. The refractive index of the surrounding medium (air) was set to be 1.0. The dimensions of the air layer and the Al₂O₃ substrate in the simulation region were made to be larger than the three times of the incident light wavelength. When we used the COMSOL software for numerical simulation, free tetrahedral meshes were employed in the simulation region while cuboid meshes were used in the perfectly matched layer. In comparison, we used Yee meshes in the FDTD simulations. In

order to obtain accurate results, the maximum mesh size was set to be 1.0 nm in both cases. The electric and magnetic field distributions in the Si/SiO₂ cuboid were extracted from the field detectors inserted in it. The Q factor of the Si/SiO₂ cuboid was extracted by monitoring the electric field decay inside the Si/SiO₂ cuboid or by fitting the scattering spectrum of the Si/SiO₂ cuboid (see Supplementary Note 5.1 and 5.2).

The multipolar expansion method⁴² was employed to decompose the total scattering of a Si/SiO₂ cuboid into the contributions of electric and magnetic modes of different orders, including ED, MD, EQ, MQ, EO, and MO etc.”, see 6th paragraph in page 12). In addition, we have added a reference for the multipolar expansion method (see Ref. 42).

C3: I am confused by the definition of external quantum efficiency noted in the SI: “the number of emitting photons to that of detected photons”. This strikes me as non-standard, and perhaps I am misunderstanding it. What do you mean by "detected photons"? I was expecting to see the EQE characterized as the ratio of emitted photons to photons impinging on the device. Additionally, it would be worth revisiting the literature cited in the table, for accuracy. For instance, browsing a random paper from the table, Ref. 7, it seems to me that what they report are internal quantum efficiency, not external.

R3: We would like to thank the reviewer for carefully reading the manuscript and supplementary materials. The reviewer is right. The external quantum efficiency is defined as the ratio of the number of photons emitted from a material to the number of exciting photons absorbed by the material. We are sorry for the mistake. We have corrected this definition in the revised version. (“Apart from the internal quantum efficiency, the external quantum efficiency, which is defined as the ratio of the number of photons emitted from a material to the number of exciting photons absorbed by the material, is also used to characterize the photoluminescence from a material.” see 1st paragraph in page 23 in Supplementary Materials).

In Ref. 7, the quantum efficiencies of amorphous Si and Si nanocrystals were measured by using an integrating sphere. Therefore, we think that the quantum efficiencies obtained by the authors belong to external quantum efficiencies. The internal (or intrinsic) quantum

efficiencies were derived based on some assumptions. According to the suggestion of the reviewer, we have revisited the accuracies of the quantum efficiencies reported in literature, as shown in the following. For the convenience of reviewing, we have copied and pasted the sentences that describe the quantum efficiencies in the reference papers. Moreover, we have corrected some errors in literature survey in the revised version (see Supplementary Note 11).

Table T1: Internal and external quantum efficiency of silicon-based materials

Materials	Preparation method	Quantum efficiency	Measurement method	Reference
Porous Si (for most samples)	Anodization and stain etching	1%–10%	External quantum efficiency	Ref. (4) ⁴ (REVIEW)
odic oxidation at room temperature. Estimated external quantum efficiency (EQE) lies in the range 1%–10%. ^{360,361}				
Single porous Si nanoparticles	Anodization and stain etching	88% (for single) 2.8% (for total)	External quantum efficiency	Ref. (5) ⁵
If we assume that the emission from porous Si samples comes only from these high QE nanoparticles than (88% QE/nanoparticle)(2.8% luminescent nanoparticles/total nanoparticles)=2.5% represents an upper bound for the QE of bulk porous Si. Previous measurements of the QE of bulk				
Amorphous Si quantum dots	plasma-enhanced chemical vapor deposition (PECVD)	2×10^{-3} %	External quantum efficiency	Ref. (6) ⁶
An external quantum efficiency of 2×10^{-3}%				
Amorphous Si nanoparticles	nonthermal low-pressure plasma reactor	1, 2% 2, 45%	1, External quantum efficiency 2, Internal quantum efficiency	Ref. (7) ⁷
quantum yield of amorphous nanoparticles is 2%, detect fraction of NCs. For instance, if a-NPs were nonemitting, a fraction of 1% of nanocrystals with an intrinsic QY of 45% would be interpreted as a sample with an ensemble quantum yield of 0.45%, a value which is consistent with the				
Crystalline Si nanoparticles	nonthermal low-pressure plasma reactor	>40%	External quantum efficiency	Ref. (7) ⁷
gest that the upper bound for the quantum yield of amorphous nanoparticles is 2% , while the quantum yield of silicon nanocrystals is routinely found to exceed 40% .				

Si quantum dots	wet-chemical method	4%	External quantum efficiency	Ref. (8) ⁸
Supplementary Information). The absolute quantum efficiency has been measured using an integration sphere technique, following the method described in Refs. 27 and 28. Using the experimentally obtained values $\tau_{\text{eff}} \approx 3\text{--}4$ ns and $\eta_{\text{ext}} \approx 4\%$, we arrive to $k_{\text{rad}} \approx 10^7$ s ⁻¹				
Si quantum Dots	wet-chemical method	3.7%–11%	External quantum efficiency	Ref. (9) ⁹
peak intensity (dashed lines). In the first case, the integrated PL emission is roughly proportional to the external quantum efficiency (QE). The absolute QE under 370 nm excitation has been estimated to be ~3.7% (for details, see SI). Therefore, it follows that the QE under 320 nm excitation would be <1% and at 405 nm around ~11%. The QE increases with				
Si nanocrystals separated by SiO ₂ barriers	Plasma-enhanced chemical vapor deposition (PECVD)	10%–19%	External quantum efficiency	Ref. (10) ¹⁰
we are characterizing the external quantum yield (EQY). layers, we demonstrated an increase of the luminescence QY at room temperature from ~10% to ~19% for an interlayer				
Si Nanocrystals	nonthermal low-pressure plasma reaction	25%–50%	External quantum efficiency	Ref. (11) ¹¹
QY of the parent (QY ₀). Solutions of the parent in 1-dodecene/mesitylene had QY ₀ values of 25–30% (sample A, 3.5 nm mean size) and 45–50% (sample B, 4 nm mean size) immediately after synthesis. The QY is For the solutions, absolute QY was measured using a fiber-coupled 395 nm LED and an integrating sphere. Where indicated, measure-				
Si nanocrystals	nonthermal low-pressure plasma reaction	<62%	External quantum efficiency	Ref. (12) ¹²
ganic ligands under strict exclusion of oxygen. High quantum yields of up to 62% at wavelengths of ~789 nm were observed. This quantum yield is competitive with the high				

Quantum yields of silicon quantum dots were measured using an absolute measurement. In this approach, the silicon

Si nanocrystals	non-thermal plasma reaction	60%–70%	External quantum efficiency	Ref. (13) ¹³
be relatively monodisperse and are well controllable through the plasma process parameters. Photoluminescence quantum yields as high as 60–70% have been achieved for particles luminescing in the red range of the visible spectrum.				
Silicon nanocrystals embedded in a SiO ₂	low-temperature annealing in ambients of O ₂ and H ₂	1, 9.3%–27% 2, 30%–85%	1, External quantum efficiency 2, Internal quantum efficiency	Ref. (14) ¹⁴

tively. EQEs of these samples, presented in Table I, show a similar increase from 9.3% for sample A, to 15.8% for sample B and 27% for sample C, that is, with factors of 1.7 and

074304-5 R. Limpens and T. Gregorkiewicz

J. Appl. Phys. 114, 074304 (2013)

FIG. 4. IQE as a function of emission wavelength measured for the samples with and without surface passivation. For samples A and B, IQE increases towards longer wavelengths (larger NCs), whereas for sample C (H₂ heat treatment), a constant IQE level is achieved.

Si nanocrystals	annealing dried commercial hydrogen silsesquioxane solution at 1100° C in a slightly reducing atmosphere with 5% H ₂ .	1, 30%–70% 2, 100%	1, External quantum efficiency 2, Internal quantum efficiency	Ref. (15) ¹⁵
-----------------	--	-----------------------	--	-------------------------

quantum yields of ~30–70%.

suggesting 100% internal quantum efficiency

Si nanocrystals	radio-frequency cosputtering followed by hightemperature annealing at 1100 ° C	1, 20% 2, 39%	1, External quantum efficiency 2, Internal quantum efficiency	Ref. (16) ¹⁶
-----------------	---	------------------	--	-------------------------

In the present case, with an external and average internal quantum efficiency of 20% and 39%, respectively, the fraction

Si nanocrystals	limited surface oxidation	1, 5%–43% 2, 6%–40%	1, External quantum efficiency 2, Internal quantum	Ref. (17) ¹⁷
-----------------	------------------------------	------------------------	--	-------------------------

Si nanocrystals	radiofrequency co-sputtering method	0.5%–20% 30%–90% “Step-like”	External quantum efficiency	Ref. (18) ¹⁸
-----------------	--	------------------------------------	--------------------------------	-------------------------

Figure 2 | Spectral dependence of external quantum yield of photoluminescence. Results for sample A, sample B, po-Si 1 and po-Si 2. The lower panels show multiples of the photoluminescence spectra of each sample (that is, the energy axis is multiplied by either 2 or 3, indicated by $2E_{PL}$ and $3E_{PL}$, respectively). Black dashed lines, indicating the ‘steps’, serve only as a guide to the eye.

Si nanocrystals	plasma-synthesized	12%–45%	Internal quantum efficiency (Electronic density of states (DOS) calculation by ab initio density-functional theory (DFT))	Ref. (19) ¹⁹
-----------------	--------------------	---------	--	-------------------------

and 0.01 eV (bottom). The bottom legend is the room-temperature QY, with experimental correspondence 3.2 nm (12%), 3.5 nm (20%), and 4.3 nm (45%). (Right) Measured and computed relaxation amplitude vs nanocrystal size at different T (right, 0.025 eV, dashed; 0.01 eV,

lifetime measurements

Geometry optimization and electronic structure simulations were performed with DFT in the Vienna *ab initio* Software Package (VASP). Specifically, we used the hybrid Heyd-Scuseria-Ernzerhof

Si nanocrystals	wet thermal oxidation	59% ±9%	Internal quantum efficiency (Using LDOS correction)	Ref. (20) ²⁰
-----------------	-----------------------	---------	--	-------------------------

emission rate modification yields values for the internal quantum efficiency and the intrinsic radiative decay rate of silicon nanocrystals. A photoluminescence quantum efficiency as high as 59% ±9% is found for

Si nanocrystals	co-sputtering method	40%-100%	Internal quantum efficiency (Using LDOS correction)	Ref. (21) ²¹
-----------------	----------------------	----------	--	-------------------------

For the samples annealed at 1250 °C, the estimated internal quantum efficiency is close to 100% at longer wavelength side of the PL bands, while it is saturated at 70% for that annealed at 1200 °C. The high quantum efficiency obtained

Si quantum dots (Couple to surface plasmons at the Ag/SiO ₂ interface)	annealed for 10 min at 1100 ° C	77±17 %	Internal quantum efficiency (Using LDOS correction)	Ref. (22) ²²
--	---------------------------------	---------	--	-------------------------

mine the Si QD internal quantum efficiency to be (77 ± 17)%.

Si Nanoparticles	high-pressure microdischarges	30%	External quantum efficiency	Ref. (23) ²³
------------------	-------------------------------	-----	-----------------------------	-------------------------

Reported values for the external quantum efficiency of np-Si have ranged from less than 1% to as high as 23%.^{2,3,5,8}

of data, shown in Figure 7, suggests that the quantum efficiency of the octanol-capped Si nanoparticles was 30%.

REVIEWERS' COMMENTS

Reviewer #2 (Remarks to the Author):

I would like to thank the authors again for their kind consideration of my comments. As in the previous round, I think that the manuscript is ready for publication.

I will simply suggest the authors to perhaps also consider defining internal quantum efficiency in the manuscript, so that it is clear what they are referring to when classifying the literature in the table. For instance, from my perspective I wouldn't quite know what to understand in that difference, as these are the definitions that I expect when talking about quantum efficiencies (mostly in relationship with the usage of the term in solar cells, of which I am more familiar than photoluminescence):

InternalQE is the ratio of emitted photons to **absorbed** photons.

ExternalQE is the ratio of emitted photons to photons **impinging** on the sample.

Consequently, the definition given in the manuscript for EQE would correspond with what I would understand to be IQE or QY. Being explicit about both definitions would easily avoid confusing readers with the same perspective as mine.

Responses to the comments of the reviewer #2

(Manuscript ID: NCOMMS-21-32514C)

We would like to thank the effort made by the reviewer for improving the quality of the manuscript. The responses (abbreviated as R) to the comments/suggestions (abbreviated as C) of the reviewer and the changes we made in the revised manuscript (marked in red color) are described in the following.

C1: I would like to thank the authors again for their kind consideration of my comments. As in the previous round, I think that the manuscript is ready for publication.

R1: We would like to thank the reviewer for the recommendation.

C2: I will simply suggest the authors to perhaps also consider defining internal quantum efficiency in the manuscript, so that it is clear what they are referring to when classifying the literature in the table. For instance, from my perspective I wouldn't quite know what to understand in that difference, as these are the definitions that I expect when talking about quantum efficiencies (mostly in relationship with the usage of the term in solar cells, of which I am more familiar than photoluminescence): Internal QE is the ratio of emitted photons to *absorbed* photons. External QE is the ratio of emitted photons to photons *impinging* on the sample. Consequently, the definition given in the manuscript for EQE would correspond with what I would understand to be IQE or QY. Being explicit about both definitions would easily avoid confusing readers with the same perspective as mine.

R2: We would like to thank the reviewer for this suggestion, which may avoid any misunderstanding for the readers in different research fields. When talking about quantum efficiency, one should distinguish two types of materials/devices. For a solar cell or a photodetector in which absorbed photons are converted into charge carriers (or electron-hole pairs), the internal quantum efficiency (IQE) and external quantum efficiency (EQE) are defined as follows:

$$\text{IQE} = \frac{\text{(the number of charge carriers collected by the solar cell)}}{\text{(the number of photons absorbed by the solar cell)}} \quad (1)$$

$$\text{EQE} = (\text{the number of charge carriers collected by the solar cell})/(\text{the number of photons shining on the solar cell}) \quad (2)$$

Here, we present the definitions of IQE and EQE for solar cells taken from Wikipedia in the following. Therefore, the definitions of IQE and EQE given by the reviewer are suitable for solar cells and photodectors. In this case, IQE is larger than EQE because only a fraction of photons shining or impinging on the solar cell (sample) is absorbed.

Types [edit]

Two types of quantum efficiency of a solar cell are often considered:

- **External Quantum Efficiency (EQE)** is the ratio of the number of charge carriers collected by the solar cell to the number of photons of a given energy *shining on the solar cell from outside* (incident photons).
- **Internal Quantum Efficiency (IQE)** is the ratio of the number of charge carriers collected by the solar cell to the number of photons of a given energy that shine on the solar cell from outside *and* are absorbed by the cell.

The IQE is always larger than the EQE in the visible spectrum. A low IQE indicates that the active layer of the solar cell is unable to make good use of the photons, most likely due to poor carrier collection efficiency. To measure the IQE, one first measures the EQE of the solar device, then measures its transmission and reflection, and combines these data to infer the IQE.

$$\text{EQE} = \frac{\text{electrons/sec}}{\text{photons/sec}} = \frac{(\text{current})/(\text{charge of one electron})}{(\text{total power of photons})/(\text{energy of one photon})}$$

$$\text{IQE} = \frac{\text{electrons/sec}}{\text{absorbed photons/sec}} = \frac{\text{EQE}}{1-\text{Reflection-Transmission}}$$

The external quantum efficiency therefore depends on both the absorption of light and the collection of charges. Once a photon has been absorbed and has generated an electron-hole pair, these charges must be separated and collected at the junction. A "good" material avoids charge recombination. Charge recombination causes a drop in the external quantum efficiency.

Copied from Wikipedia. https://en.wikipedia.ahmu.cf/wiki/Quantum_efficiency

When dealing with a light emitting device (LED) which converts injected charge carriers into photons, however, the IQE and EQE of the LED are generally defined as follows¹:

$$\text{IQE} = (\text{the number of photons produced within the device})/(\text{the number of injected charge carriers (or electron-hole pairs)}) \quad (3)$$

$$\text{EQE} = (\text{the number of photons emitted externally from the device})/(\text{the number of injected charge carriers (or electron-hole pairs)}) \quad (4)$$

For the convenience of reviewing, we have copied the definitions of IQE and EQE for a LED from Ref. 1 in the following. Still, the IQE of the LED is also larger than the EQE because the photons produced within the device may not be able to emit from the device into free space.

For LEDs, the EQE can be evaluated from the ratio of collected EL photon number per second to the injected electron number. The internal quantum efficiency (IQE, η_{int}) and EQE (η_{ext}) can be calculated using the following equation:

$$\eta_{int} = \frac{P_{int}/h\nu}{I/e} \quad (6)$$

$$\eta_{extraction} = \frac{P/h\nu}{P_{int}/h\nu} \quad (7)$$

$$\eta_{ext} = \frac{P/h\nu}{I/e} = \eta_{int}\eta_{extraction} \quad (8)$$

where P_{int} denotes the optical power emitted from the active region and can be simplified to the number of photons emitted from the active region per second. P denotes the optical power

Copied from Ref. 1

Similar to LEDs, the IQE and EQE of a luminescent material excited by light (i.e., the photoluminescence) are commonly defined as follows²:

IQE = (the number of photons generated inside the material)/(the number of photons absorbed by the material) (5)

EQE = (the number of photons emitted out of the material)/ (the number of photons absorbed by the material) (6)

For the convenience of reviewing, we have copied the definitions of IQE and EQE for a luminescent material from Ref. 2 in the following.

As we know that PL IQE (ε) is defined as the ratio of the number of photons generated inside the sample ($\phi_{int}(\lambda_{emi})$) over the number of absorbed photons at an excitation wavelength ($\phi_{abs}(\lambda_{exc})$), while the PL EQE (η) is defined as the ratio of the number of photons emitted out of the sample ($\phi_{ext}(\lambda_{emi})$) over $\phi_{abs}(\lambda_{exc})$. The relationship between ε and η can be given by $\varepsilon = \eta/N^*$, where N^* is called light extraction

Copied from Ref. 2

In general, only the EQE of a device/material can be experimentally determined while the corresponding IQE is derived from the EQE if the extraction efficiency of photons is known. In this work, the Si/SiO₂ cuboids belong to luminescent materials whose IQE and EQE are characterized by using Eqs. (5) and (6). In our experiments, we derived the EQE of a Si/SiO₂ cuboid by measuring the number of photons emitted from the Si/SiO₂ cuboid and the number of photons absorbed by it (see Supplementary Note 12 for a detailed description). According to the suggestion of the reviewer, we have added the definitions of IQE and EQE for Si/SiO₂ cuboids in the main text and made it clear that the quantum efficiency reported in this work is the EQE of Si/SiO₂ cuboids (“In this way, the **external** quantum efficiency for the luminescence of the Si/SiO₂ cuboid, **which is defined as the ratio of the number of photons emitted out of the Si/SiO₂ cuboid to the number of absorbed photons**, is found to be ~13% (Supplementary Note 12). **This value** is further improved when comparing with the previous results for Si nanoparticles on an Ag film²². The internal quantum efficiency, which is given by **the ratio of the number of photons generated inside the Si/SiO₂ cuboid to the number of absorbed photons, should be larger than the external quantum efficiency**”, see the last paragraph in page 9).

We would like to thank the reviewer again for his/her effort to improve the quality of the manuscript. We hope that the reviewer will be satisfied with the revised manuscript.

References

1. Caiyun W. *et al.* The highly-efficient light-emitting diodes based on transition metal dichalcogenides: from architecture to performance. *Nanoscale Adv.* **2**, 4323 (2020).
2. Pengzhan Z. *et al.* Higher than 60% internal quantum efficiency of photoluminescence from amorphous silicon oxynitride thin films at wavelength of 470 nm. *Appl. Phys. Lett.* **105**, 011113 (2014).